

# Insensitivity of mass loss of Icelandic Vatnajökull ice cap to solar geoengineering

Chao Yue[1], Louise Steffensen Schmidt[2], Liyun Zhao[1,3], Michael Wolovick[1,4], John C. Moore[1,5,6]

[1]College of Global Change and Earth System Science, Beijing Normal University, 100875 Beijing, China
[2]Department of Geosciences, University of Oslo, 0316 Oslo, Norway
[3]Southern Marine Science and Engineering Guangdong Laboratory, 519000 Zhuhai, China
[4]Alfred Wegener Institute, 27570 Bremerhaven, Germany
[5]CAS Center for Excellence in Tibetan Plateau Earth Sciences, 100101 Beijing, China
[6]Arctic Centre, University of Lapland, 96101 Rovaniemi, Finland

*Correspondence to*: Liyun Zhao (zhaoliyun@bnu.edu.cn), John C. Moore (john.moore.bnu@gmail.com)

**Abstract.** Geoengineering by stratospheric aerosol injection (SAI) may reduce the mass loss from Vatnajökull ice cap (VIC), Iceland, by slowing surface temperature rise, despite relative increases in ocean heat flux brought by the Atlantic Meridional Circulation (AMOC). Although surface mass balance (SMB) is affected by the local climate, the sea level contribution is also dependent on ice dynamics. We use the Parallel Ice Sheet Model (PISM) to estimate the VIC mass balance under the CMIP5
(Coupled Model Intercomparison Project Phase 5) RCP4.5, 8.5 and GeoMIP (Geoengineering Model Intercomparison Project) G4 SAI scenarios during the period 1982-2089. The G4 scenario is based on the RCP4.5, but with additional 5 Tg yr$^{-1}$ of SO$_2$ injection to the lower stratosphere. By 2089, G4 reduces VIC mass loss from 16 % lost under RCP4.5, to 12 %. Ice dynamics are important for ice cap loss rates, increasing mass loss for RCP4.5 and G4 by 1/4 to 1/3 compared with excluding ice dynamics, but making no difference to mass loss difference under the scenarios. We find that VIC dynamics are remarkably
insensitive to climate forcing partly because of AMOC compensation to SMB and low rates of iceberg calving making ocean forcing close to negligible. But the exceptionally high geothermal heat flow under parts of the ice cap which produces correspondingly high basal melt rates means that surface forcing changes are relatively less important than for glaciers with lower geothermal heat flow.

## 1 Introduction

Iceland in the northern North Atlantic is strongly affected by the warm Irminger and cold East Iceland ocean currents, leading to a relatively wet, oceanic climate with high precipitation, mild winters around 0 ℃ and a mean annual temperature of about 5℃ near the southern coast (Einarsson, 1984). The unique climate has allowed 11% of the country to be covered by glaciers (Björnsson and Pálsson, 2008). Glaciers in Iceland are temperate, with a high annual mass turnover having both high accumulation and high melt rates, which since maritime glaciers are more sensitive to climate variations than continental ones,
might be expected to make Icelandic glaciers bellwethers of glacier response to climate warming. Indeed, as the climate has warmed, glaciers in Iceland have experienced considerable mass loss in recent decades, with melting expected to accelerate



throughout this century (Schmidt et al., 2017; Schmidt et al., 2020; Aðalgeirsdóttir et al., 2020). Although their contribution to global mean sea-level rise would be just 1 cm, even if all the ice melted (Björnsson and Pálsson, 2008), the local impacts of rapid glacier loss will be obvious and deeply moving for Icelanders and cause profound changes in hydrology.


Vatnajökull ice cap (VIC), the largest nonpolar ice cap in Europe with a volume of 2870 km$^3$ and an area of 7700 km$^2$ (Aðalgeirsdóttir et al., 2020), is shrinking at an increasing rate (e.g., Björnsson et al., 2013). Surface mass balance (SMB, the sum of accumulation and ablation) significantly decreased from a slightly positive balance in the 1980s to -0.8 m yr$^{-1}$ during 1995-2014 (Pálsson et al., 2017). Projections show that VIC volume will decrease 51-94 % by the year 2300 under the

representative concentration pathway (RCP) 8.5 scenario (Schmidt et al., 2020).

Geoengineering by stratospheric aerosol injection (SAI) may help to meet IPCC (Intergovernmental Panel on Climate Change) targets of limiting global mean temperature rises to less than 2°C, since present greenhouse gas emissions trajectories suggest temperatures over much of the latter half of the 21st century will exceed this limit (MacMartin et al., 2018). Moreover, SAI

may be a relatively cheap way to offset temperature rises on the global scale (Smith and Wagner, 2019). SAI does reduce surface runoff over VIC compared with RCP4.5 and RCP8.5 (Yue et al., 2021), but by relatively little compared with the SAI induced reductions in mass loss from both Greenland (Moore et al., 2019) and smaller mountain glaciers (Zhao et al., 2017). However, Yue et al. (2021) did not consider non-surface mass balance generated by changes in ice flow and discharge (e.g., calving of ice and basal melting) that are driven by changing climate or impacts due to the warming ocean in contact with the

ice. It is this component that we tackle here. Although there are many other potential impacts that might be expected if SAI were ever undertaken, here we focus only on the mass balance of a single ice cap in Iceland. While this is a very narrow focus, the study is topical and of wider relevance because of the atypical behaviour of the North Atlantic under both greenhouse gas forcing and SAI, primarily because of the compensatory effects the climate forcing has on the AMOC. The AMOC is simulated as slowing considerably under greenhouse gas forcing (Cheng et al., 2013), while slowing much less under geoengineering

(Hong et al., 2017; Yue et al., 2021). Furthermore, the Arctic is currently warming at least twice as fast as the global mean temperature, leading to concerns on the stability of the Arctic cryosphere and examination of possible roles for geoengineering methods its preservation (Lee et al., 2021). Thus, the North Atlantic Arctic sector is a key region for examination of unwelcome impacts from geoengineering.

We simulate the response of the VIC with the state-of-the-art Parallel Ice Sheet Model (PISM) driven by monthly surface mass balance  from 2006-2089 under SAI and greenhouse gas climates. The RCP4.5 emissions scenario (Thomson et al., 2011) is close to future emissions under the 2015 Paris climate agreement (Kitous and Keramidas, 2015), while the RCP8.5 scenario (Riahi et al., 2011) is a "business-as-usual" extreme failure to mitigate scenario. The SAI G4 scenario branches off the RCP4.5 scenario at 2020, specifying 5 Tg yr$^{-1}$ of SO$_2$ to be injected into the equatorial lower stratosphere until 2069, and then continues

with RCP4.5 forcing to 2089 (Kravitz et al., 2013).



## 2 Model and Verification

The PISM model (version 1.0; Bueler and Brown (2009); https://pism-docs.org/wiki/doku.php) is an open-source ice sheet thermo-dynamic model that has been used in numerous studies of a wide range of ice sheets and glaciers (e.g., Aschwanden et al., 2019; Yan et al., 2020). PISM includes a hybrid stress balance model (Bueler and Brown, 2009) with both Shallow Ice

Approximation (SIA; Hutter, 1983) and Shallow Shelf Approximation (SSA; Morland, 1987) to solve ice vertical deformation and longitudinal stretching allowing simulation of both slowly flowing ice cap interiors and fast flowing outlet glaciers. Schmidt et al. (2020) optimized the free parameters in PISM using observations over Vatnajökull, and we use these choices for all our simulations, with the SIA+SSA hybrid schemes for ice stress balance, the isothermal Glen flow law for ice flow, the pseudo-plastic flow law for ice basal sliding, and the Eigen scheme for ice calving.


To initialize PISM over the VIC, we need the boundary conditions of the surface elevation, bedrock altitude, ice thickness, upward geothermal flux, ice temperature, and surface mass balance (Fig. 1). We re-grided these to 500 m×500 m resolution. As with Schmidt et al. (2020), we prescribe temperate ice at 0°C for the whole VIC. We use the daily SMB field (Yue et al., 2021) from the surface energy and mass balance model of intermediate complexity, SEMIC, (Krapp et al., 2017) under the

historical, G4, RCP4.5 and RCP8.5 scenarios during 1982-2089 driven by the four ESMs with data available: BNU-ESM (Ji et al., 2014), HadGEM2-ES (Collins et al., 2011), MIROC-ESM (Watanabe et al., 2010), and MIROC-ESM-CHEM (Watanabe et al., 2010). These ESMs were statistically downscaled and bias corrected using ISI-MIP (Hempel et al., 2013) and lapse rate approaches to achieve a spatial resolution of 0.025°×0.025°. The derived SMB (Fig. S1) is in reasonable agreement with observations over VIC.





**Figure 1.** Model input data fields. (a) Vatnajökull ice cap location map with surface elevation from Spot5 (Berthier and Toutin, 2008) in summer 2010; (b) bedrock elevation from radar mapping in 1980 (Björnsson, 1986, 1988); (c) ice thickness; (d) average surface mass balance 1982-2005 simulated by SEMIC forced by four ESMs, (Yue et al., 2021), with the black dashed curve marking the equilibrium line boundary between net accumulation and net ablation; (e) upward geothermal heat flux (Flowers et al. 2003), including the Grímsvötn active volcano.

We drive the PISM model with monthly SMB fields from four ESMs during 1982-1999, when the ice cap was close to steady-state, repeated for 2000 years (Fig. 2), achieving a near steady-state ice cap geometry. The final year of spin-up is then used as the initial condition in simulations.





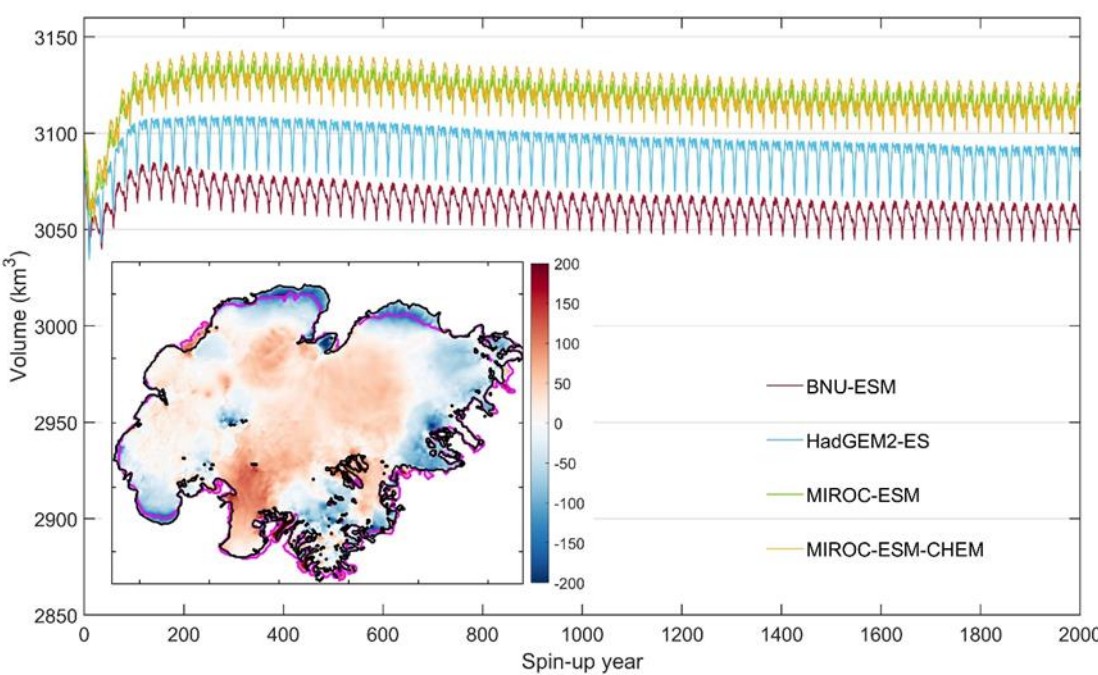

**Figure 2.** Changes in Vatnajökull ice cap volume from the 2000-year climate spin-up driven by repeated 1982-1999 SMB fields from PISM forced with BNU-ESM, HadGEM2-ES, MIROC-ESM and MIROC-ESM-CHEM. The equilibrium volume is slightly different than present day by -1.3% for BNU-ESM, -0.5% for HadGEM2-ES, and 0.8% for both MIROC models. Lower left corner subplot is the ensemble mean of spatial distribution of VIC thickness differences (ice thickness after spin-up minus present ice thickness) from PISM. The black curves represent the present ice cap extent. The magenta curves represent the extent after spin-up.

The SMB-altitude feedback alters SMB as VIC topography evolves. VIC surface elevation and historical SMB over 1982-2005 are significantly correlated ($R^2 > 0.7$, $p < 0.01$). We therefore correct the annual SMB forcing with the ESM-dependent "SMB lapse rate" $k$, and the ice thickness change modeled by PISM in year t from 2006 to 2089:

$$SMB_t^{adj} = SMB_t^{SEMIC} + k \times (h_{t-1}^{PISM} - h_0^{PISM})$$

Where $SMB_t^{adj}$ and $SMB_t^{SEMIC}$ are the corrected and SEMIC modeled SMB in year $t$, $h_{t-1}^{PISM}$ is the ice thickness in the year $t$-$1$ modeled by PISM, and $h_0^{PISM}$ is the ice thickness in 2005. The values of $k$ in Eqn. (1) are {4.59, 4.34, 4.28, 4.28} mm m$^{-1}$ for BNU-ESM, HadGEM2-ES, MIROC-ESM and MIROC-ESM-CHEM, respectively.

During spin-up, VIC volume changes by between -1.3% and 0.8% for the four ESMs relative to present day, while area is reduced by around 16% (Fig. 2). Ice area loss is mainly over the outlet glaciers of Dyngjujökull, Brúarjökull and Síðujökull



(Figs. 1, 2) where the ice thicknesses are less than 100 m. Changes in VIC geometry are largely determined by the SMB field and are consistent across all the ESMs because their climate variables (e.g., surface air temperature, downward longwave and shortwave radiation) that drive SMB are bias-corrected with ERA5 reanalysis. This tends to impose a fixed spatial pattern to SMB while preserving the long-term trends in the ESM forcing fields, which are small during the 1982-1999 period. The largest discrepancies between the spin-up and present area for VIC, especially the reductions over the northern glaciers and

increased thickness over Skeiðarárjökull, are likely caused by the quasi-periodic surging glaciers that are not parameterized by PISM, but which cover about 75% of the VIC (Björnsson et al., 2003). VIC surges occur on timescales from several years to centuries, and rapidly move accumulated ice from the upper glacier towards the terminus, meaning that the upper and lower glacier are usually out of steady state. Thus, the spin-up is unlikely to achieve a present-day area coverage, although total volume is close to observed.


We next compare the present-day VIC volume and ice surface velocity to observations. Pálsson et al. (2015) record a 3% reduction in volume between 1991-2014, which is more than we simulate (Table 1), partly due to the slight overestimation of VIC SMB used to force PISM compared with measurements (Schmidt et al., 2020; Yue et al., 2021), and the disappearance of fast melting regions. Sentinel-1 satellite observed and PISM modeled VIC ice velocity (Fig. 3) show very similar spatial

patterns during 2015-2020, with a maximum of >350 m yr$^{-1}$ over Skeiðarárjökull and Breiðamerkurjökull, and relatively lower velocities of <100 m yr$^{-1}$ over almost the whole northern VIC. However, there are some large differences mainly over the eastern outlet glaciers where PISM overestimates the velocity by more than 100 m yr$^{-1}$. The likely reason is that spin-up produces a somewhat different ice cap geometry from observed (Fig. 2).



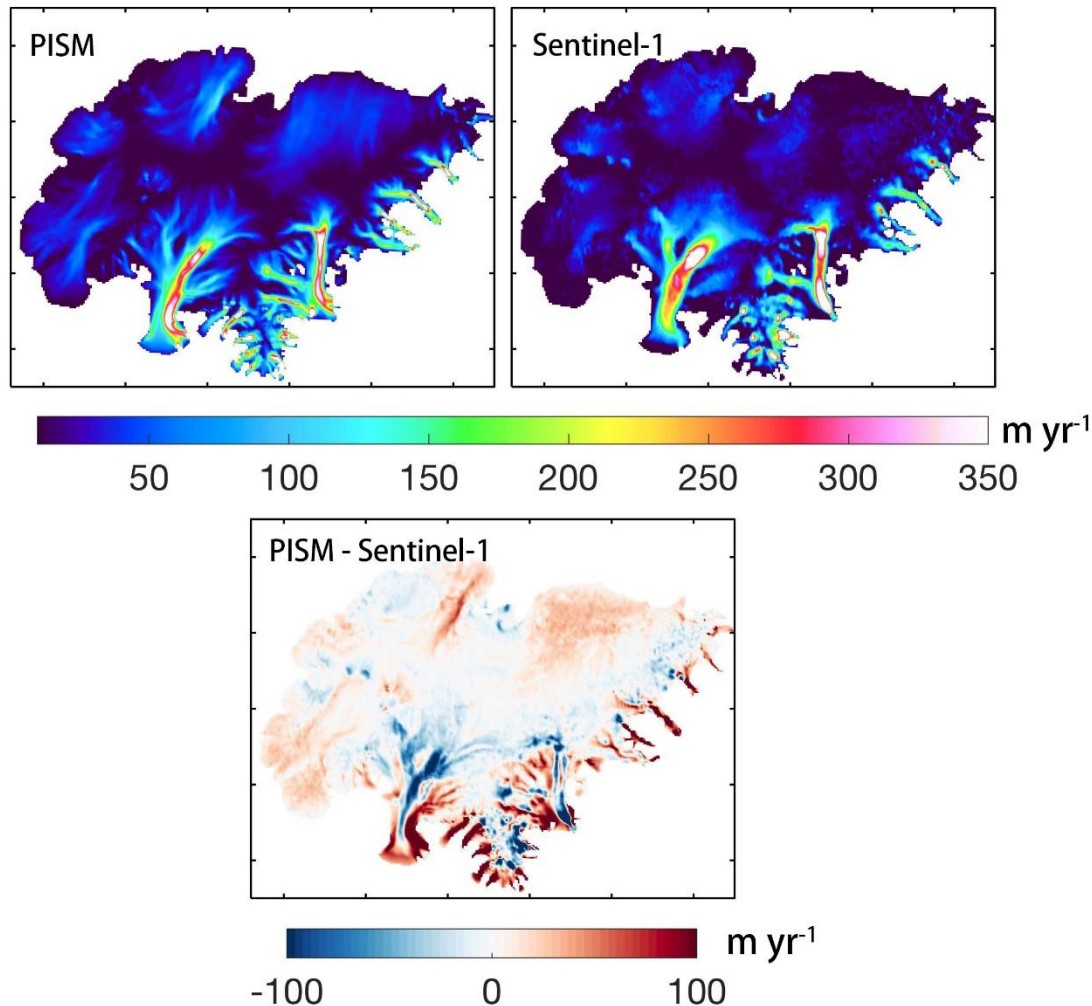

**Figure 3.** Across-ESM ensemble and RCP4.5 and RCP8.5 scenario average spatial distribution of mean surface velocity over VIC during 2015-2020 (upper left; calculated by the mean of surface velocity under RCP4.5 and RCP8.5), satellite observation from Sentinel-1 100 m spaced product (Wuite et al., 2021) (upper left), and their difference (bottom): PISM output-Sentinel.

## 3 Ice cap volume and area from 1982 to 2089

Table 1 and Fig. 4, shows how the simulated VIC volume slightly increased, while area reduced during the historical period 1982-2005, and then decreases under all future scenarios along with the decreased SMB (Fig. S1). By the end of the year 2089, the across-ESM ensemble mean of VIC volume is decreased by as little as 12% under G4 to as much as 22% under RCP8.5 (Table 1). G4 reduces the VIC volume 7-8% relative to RCP4.5 with the BNU-ESM and HadGEM2-ES forcing, but the two MIROC models predict little differences, due to their statistically insignificant differences of SMB in these scenarios. By 2089, all four ESMs simulations produce surface thinning over the whole VIC especially over Tungnaarjökull, Brúarjökull and





eastern small outlet glaciers (Fig. 5). Surface thinning under G4 is smaller than that under RCP4.5 and RCP8.5. G4 increases
ensemble ice thickness by around 10-20 m relative to RCP4.5, and 40-70 m relative to RCP8.5.



**Figure 4.** Time series of Vatnajökull ice cap volume (upper) and area (bottom) change during 1982-2089 driven by (from left
to right) BNU-ESM, HadGEM2-ES, MIROC-ESM, MIROC-ESM-CHEM and ensemble mean under historical (magenta), G4
(red), RCP4.5 (blue) and RCP8.5 (black) scenarios. Estimates considering ice dynamic from PISM are shown in solid curves,
with volume change only caused by SMB as dashed curves. Vertical dashed lines mark the beginning and end of the SAI
geoengineering.



**Figure 5.** The ice thickness differences from PISM outputs between the year 2089 and 1982 over Vatnajökull ice cap under
G4 (1st row), RCP4.5 (2nd row) and RCP8.5 (3rd row) scenarios, and their differences (G4-RCP4.5, 4th row; G4-RCP8.5, 5th
row) by (from left to right), BNU-ESM, HadGEM2-ES, MIROC-ESM, MIROC-ESM-CHEM and ensemble mean.



**Table 1.** Vatnajökull ice cap volume and area change (%) by 2089 relative to 1982 under G4, RCP4.5 and RCP8.5 scenarios modeled by PISM forced by BNU-ESM, HadGEM2-ES, MIROC-ESM, MIROC-ESM-CHEM, the ensemble mean and 95% confidence intervals, N=4 Numbers in brackets represent changes without considering SMB-elevation feedback.

|  |  | BNU-ESM | HadGEM2-ES | MIROC-ESM | MIROC-ESM-CHEM | Ensemble |
|---|---|---|---|---|---|---|
| Volume | G4 | 14 (13) | 10 (10) | 11 (11) | 13 (12) | 12±2 |
|  | RCP4.5 | 21 (20) | 18 (17) | 11 (11) | 13 (13) | 16±4 |
|  | RCP8.5 | 25 (23) | 23 (22) | 20 (20) | 22 (21) | 22±2 |
|  | 1991-2014 | 2 | 2 | 0 | 0 | 1±1 |
| Area | G4 | 10 (9) | 6 (6) | 8 (7) | 9 (8) | 8±2 |
|  | RCP4.5 | 14 (12) | 11 (11) | 8 (7) | 9 (9) | 10±3 |
|  | RCP8.5 | 15 (14) | 14 (14) | 12 (12) | 13 (12) | 14±1 |

## 4 Ice cap SMB, MB and non-SMB from 2020-2089

In Fig. 6 we separate the SMB and non-SMB (ice dynamics and basal melting) components of overall mass balance. SMB increases the ice thickness over the interior of VIC, with the maximum of more than 400 m over the southern region of VIC, while decreasing the ice thickness over the margins, especially over the outlet glaciers of Skeiðarárjökull and Breiðamerkurjökull. The degree of surface thinning under RCP8.5 is stronger than for the RCP4.5 and G4 scenarios, resulting in the smallest area of surface thickening. Non-SMB components display the opposite pattern to SMB. Positive non-SMB contributions are visible in all ESMs and scenarios over the margins, because as the negative SMB steepens the margins, it is compensated by increased ice flow from the interior and so dynamically thickens. Similarly, the drawdown of ice from the interior thins the higher elevation ice cap making the non-SMB component there negative. Basal melting is driven by non-climate factors and so remains essentially unchanged under the scenarios. The pattern of non-SMB contributions for individual ESM are all quite similar, the largest differences being mainly over the ablation zone, with ensemble standard deviations of more than 10 m (Fig. S2). Fig. 6b demonstrates that surface height differences (G4-RCP4.5 and G4-RCP8.5) by 2089 are mainly caused by SMB rather than dynamic effects, SMB under G4 increases VIC mean surface height by around 20 m. SAI dynamically thickens the ablation zone relative to the RCP scenarios, while thinning the accumulation area. The dynamic impact on surface height change differences between G4 and RCP4.5 are confined to ±10 m, which is much less than differences from RCP8.5.

Fig. 6c illustrates the across-ESM mean time series of modeled MB, SMB and non-SMB. MB is well-correlated ($R^2$=0.98, p<0.01) with SMB, and generally exhibits a downward trend in all ESMs and scenarios, although with high annual variability. The non-SMB contributions, however, remain nearly constant (around -0.25 m yr$^{-1}$) over time and across all scenarios (Fig. 7). By 2069, ice dynamics are important for ice cap loss rates, increasing mass loss for RCP4.5 and G4 by 1/4 to 1/3 compared with excluding dynamics, but making no difference to mass loss difference under the scenarios.





**Figure 6.** a) Ensemble mean of ice cap height differences, 2089 minus 2020 caused by SMB (first row) and non-SMB (i.e.,

ice dynamics and basal melting, (second row)) calculated as the difference between MB and SMB, under the G4, RCP4.5 and

RCP8.5 scenarios. No change is marked by the dashed black curves. b) Ensemble mean differences (G4-RCP4.5 and G4-

RCP8.5) in ice cap thickness by 2089 due to SMB (first row) and ice dynamics (second row). c) Decadal ensemble means of

modeled mass balance (solid curves), SMB (dashed curves) and non-SMB (dotted curves) under historical (magenta), G4 (red),

RCP4.5 (blue) and RCP8.5 (green) scenarios. The vertical lines denote the beginning and the end of SAI geoengineering.

Individual models are in Figs. S3-S6.





SMB gradually declines, and accounts for 52±25% (95% confidence intervals; N=4) changes of MB (Figure 2c) during the SAI period 2020-2069 in G4. G4 increases MB by 0.21±0.17 m yr$^{-1}$ and 0.33±0.22 m yr$^{-1}$ (95% confidence intervals; N=4) compared with RCP4.5 and RCP8.5, although two MIROC models project almost no MB differences between G4 and RCP4.5. The mean SMB and MB under G4 have much larger differences between the ESMs than the two RCP scenarios (Fig. 7).



**Figure 7.** The mean MB, SMB, and non-SMB (MB-SMB) over VIC during 2020-2069 under G4, RCP4.5 and RCP8.5 by four ESMs and ensemble mean. Error bars represent 95% confidence intervals, N=4.



## 5 Discussion

During the historical period, our simulations show the overall mass loss on VIC is about equally divided between SMB and non-SMB components, but as SMB becomes more negative, the non-SMB fraction becomes less important. Yue et al. (2021) simulate that SAI G4 (2020-2069) could reduce surface runoff by 6.2±6% (95% confidence intervals), relative to RCP4.5.

Including the dynamic and basal melt components leads to 12±2% mass loss under G4, and 16±4% under RCP4.5 of present-day mass by 2089. So, G4 saves 25% of mass lost relative to RCP4.5, while for High Mountain Asia (HMA) it saves 19% more than RCP4.5 (Zhao et al., 2017). The differences in efficacy are related to the degree of imbalance of the ice masses to present and recent climate, with most of HMA losing ice mass throughout the last century, so losses under RCP4.5 are 73%, and under G4 59%, of present-day glacier mass. Iceland has been closer to balance until recently.


The relative effectiveness of SAI on reducing surface runoff over VIC is much smaller than for the Greenland Ice Sheet (Yue et al., 2021), largely due to the compensating impact of AMOC changes. How will this change in future when non-SMB components are also considered? Greenland ice sheet MB has been simulated by PISM under forcing from 4 ESMs simulating the RCP2.6, and RCP8.5 scenarios (Goelzer et al., 2021). AMOC declines during 2020-2089 irrespective of ESMs and

scenarios (Fig. 8a), associated with weakened deep water convection over the northern North Atlantic, but clearly, ocean overturning near Iceland is more vigorous than Greenland (Fig. 8d, e). AMOC under G4 is significantly stronger than that under RCP4.5, producing about 10 W m$^{-2}$ higher heat flux from ocean to atmosphere around Iceland than over Greenland (Yue et al., 2021). Fig. 8b-c shows that VIC MB is very significantly dependent on AMOC (R=0.91, p<0.01), while for Greenland its impact is not significant (R=0.42, p=0.35), consistent with the SMB behavior (Yue et al., 2021). Thus, AMOC plays a role

through its impact on SMB and indirectly on dynamic change through the long-term changes in ice cap geometry. Because VIC is much thinner than the Greenland ice sheet, and has higher accumulation and ablation rates, the mass turnover time in VIC is at least 10 times faster than in Greenland meaning that surface climate may induce larger dynamic effects earlier.

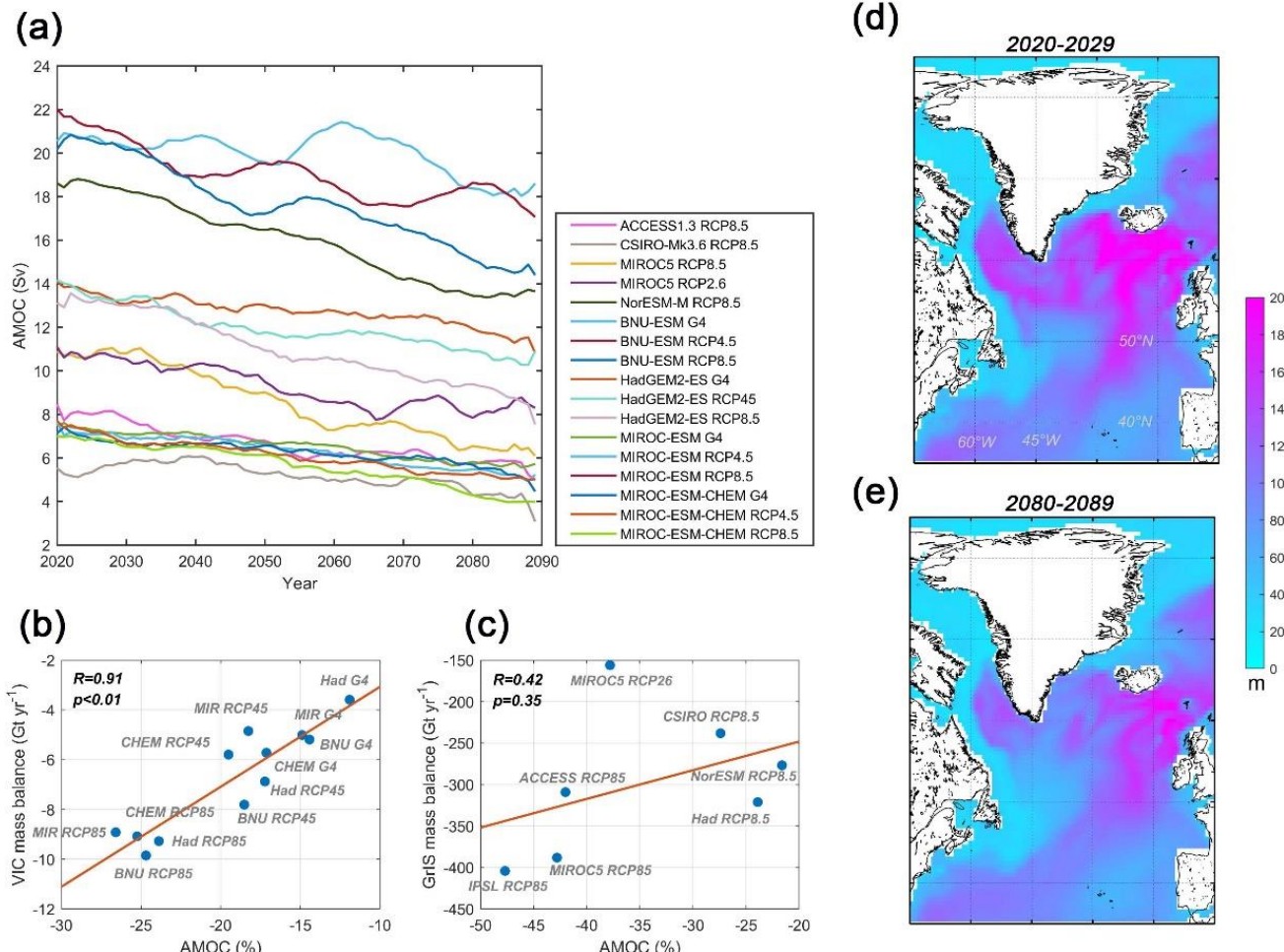

**Figure 8.** a) Decadal mean AMOC during 2020-2089 from 8 ESMs under RCP2.6, 4.5, 8.5 and G4 scenarios. b-c) Linear
regression between annual mean mass balance and relative differences of AMOC index relative to 1990-2005 over VIC (b)
and Greenland ice sheet (c) during the period 2020-2089. The ESMs and scenarios plotted are all shown in panel a). Mass loss
by PISM over Greenland is available for six ESMs for RCP8.5, but only one for RCP2.6 (Goelzer et al., 2021). The AMOC
index is calculated as the annual mean maximum volume transport stream function at 30°N. d-e) Ensemble mean (only
available for ACCESS1.3 and NorESM-M) ocean mixed layer depth during 2020-2029 (d) and 2080-2089 (e).

Instead, we find that non-SMB accounts for 0.25 m yr$^{-1}$ mass loss for all future scenarios. We find only an extra 0-2% volume
and area loss by 2089 because of the SMB-elevation feedback on VIC geometry, consistent with the 1-3% found by Schmidt
et al. (2019) who considered feedbacks based on temperature and precipitation lapse rates. The SMB-elevation feedback will
become increasingly important in longer simulations e.g., VIC volume and area will additionally reduce by 9-14% and 9-20%
by 2300 than without feedback corrections (Schmidt et al., 2020). For a period as short as the 50-year SAI application specified

in G4, the change in ice elevation feedback and the induced ice dynamical effects might be expected to be rather too small to be seen in large ice caps with a mass turnover time measured in hundreds or thousands of years. However, over Vestfonna ice cap (Svalbard), scenario-dependent impacts from 10 to 50% (RCP2.6 to RCP8.5) appeared on century timescales (Schäfer et al., 2015), and even over the large Greenland ice sheet, effects due to changing elevation-SMB and ice dynamics become

detectable (5-8%) by 2100 (Edwards et al., 2014; Goetlzer et al., 2020) and amount to 14-31% after 200 years (Goetlzer et al., 2013). Moreover, the extreme maritime environment of VIC would make it one of the glaciers most likely to exhibit a dynamical response to the SAI or RCP scenario, but we see no such effect. Perhaps the chief reason for the scenario-insensitivity is that the proportion of calving in non-SMB is small; if the calving losses at the terminus are converted to an equivalent area-average ablation rate over the upper surface of the ice cap, they only amount to 0.06 m yr$^{-1}$ (26%) over the

recent few decades (Jóhannesson et al., 2020). The remaining 74% come from basal melt by geothermal heating, volcanic eruptions, and dissipation of potential energy. Furthermore, retreat of the margins from the ocean limits ocean losses as the result of increased calving into rapidly growing terminus lake in Jökulsárlón (location see Fig. 1). Hence it is perhaps not surprising we find no trend in dynamic losses in future scenarios.


Some previous simulations of VIC had difficulty establishing present-day steady-state geometries (Aðalgeirsdóttir et al., 2005; Marshall et al., 2005; Flowers et al., 2005), and our projections are for smaller losses (16±4% for RCP4.5, and 22±2% for RCP8.5) than the e.g. 30% loss under RCP4.5 in Flowers et al. (2005). Our results are quite consistent with Schmidt et al. (2019), (17% volume loss for RCP4.5), perhaps unsurprisingly as we use the same ice dynamic model but with different SMB

forcing, which leads to local differences in various basin ice thicknesses by 2089.

The influence of volcanic deposits on snowpack albedo has not been incorporated into any snowpack simulations as yet, especially the relatively parameterized SEMIC model; in some (infrequent) years volcanic ash blankets some parts of the ice cap, drastically changing local ablation rates. The steep geometry of some outlet glaciers is still not perfectly captured by the

0.025°×0.025° grid although the bias-correction using satellite observations of albedo compensates for the resolution in the higher resolution model grid.

Of the four ESMs we utilize here, the two MIROC models share many of the same model parameterizations except for the atmospheric chemistry component coupled in MIROC-ESM-CHEM. Both models simulate little difference between G4 and

RCP4.5 over VIC. Moore et al. (2019) evaluated de-weighting in each MIROC model in Greenland simulations, but this made little difference to the ensemble means, and in general the two ESM are considered independent in climate simulations. The new generation of ESMs that participated in CMIP6, and with new corresponding GeoMIP G6 experiment are slowly becoming available and could perhaps provide improved polar impact studies.
Although geoengineering by SAI is not particularly effective for VIC, and probably other Icelandic ice caps, it does still slow
the rate of ice loss. Given the unique geographical location of the ice cap, we may infer that SAI as specified by G4 will not
lead to greater mass loss of any glacier of ice cap in the northern hemisphere than expected under any plausible greenhouse
gas scenario. However, we are not advocating that SAI be done. Arctic SAI would inevitably modify the entire global climate.
If the population of Iceland wished to make local efforts to conserve their ice caps, locally targeted interventions may offer
more palatable governance issues (Moore et al., 2020).

**6 Conclusions**

We simulate the VIC volume and area change during the period 1982-2089 using the ice sheet model PISM that forced with
monthly SMB fields calculated by four ESMs. By 2089, the SAI we simulate reduces VIC mass loss by 4 percentage points,
demonstrating that even in Iceland where the impact of AMOC is most felt, SAI could help preserve VIC from melting. The
relative unimportance of calving losses due to lack of ocean terminating glaciers, the high basal melt due to active geothermal
areas, and the compensating changes in temperature and accumulation due to AMOC mean that VIC is relatively insensitive
to climate scenario, with the ice dynamics especially so. We find that ice dynamics are almost constant over both time and
scenario because they are relatively unaffected by changing air and ocean temperatures.


*Code and data availability.* All scripts used for simulations are available upon request from the authors. The modeled ice cap
volume and area data by PISM are available at https://doi.org/10.5281/zenodo.5410852

*Author contributions.* CY, LSS, LZ, and JCM designed the experiments; CY and LSS performed them; CY, LZ, MW, and
JCM analysed the results; CY and JCM wrote the manuscript.

*Competing interests.* The authors declare that they have no conflict of interest.

*Acknowledgements.* We thank Yimeng Xu for assistance with the draft writing. We thank the climate modeling groups for
participating in the Geoengineering Model Intercomparison Project and the scientists managing the earth system grid data
nodes who have assisted with making GeoMIP output available.

*Financial support.* This study is supported by the National Key Research and Development Program of China
(2018YFC1406104), National Natural Science Foundation of China (No. 41941006), National Basic Research Program of
China (2016YFA0602701), Finnish Academy COLD consortium grant 322430.





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
