# Peer review of "Figure S1. Decadal VIC SMB modeled by SEMIC with a fixed present-day ice mask from four ESMs identified by line style (legend). Colors indicate Historical (magenta), G4 (red), RCP4.5 (blue) and RCP8.5 (black) scenarios."

_The Cryosphere, 2021_

## Referee Comment (RC1)

Review of manuscript submitted to Cryosphere: "Insensitivity of mass of Icelandic Vatnajökull ice cap to solar geoengineering" by Chao Yue, Louise Steffensen Schmidt, Liyun Zhao, Michael Wolovick and John C. Moore

General comments

This manuscript seems to me to be a follow up to a previous study by same authors published in Earth´s future (Yue et al. 2021) now with added ice flow model. The manuscript reads as uncompleted and hastily written afterthought that does not add much information to what was already published. Limited information about the models, limited understanding of ice dynamics (section 4 in particular) and poor presentation of the ensemble mean, rather than interesting results that the 4 ESM cause very different responses to the SAI, leaves reader with more questions than answers. Also, the fact that all the forcing fields are bias corrected (see comments below, some confusion about what is done) makes one wonder if any model dependent or physically caused impacts have been masked out with this bias correction and the observed responses therefore meaningless? Below are numerous comments about presentation and needs for clarifications. This manuscript needs major revisions.

Specific comments:

The title of the manuscript is misleading and even misguiding. What is "solar geoengineering"? first guess would be that some engineering is done to the sun, this phrase is not used again in the paper, but "stratospheric aerosol injection" which is not directly related to "solar geoengineering", my suggestion is to be consistent throughout the paper about what is being discussed, injection in the stratosphere is not affecting the sun, is it? Also, the mass loss of ice caps is dependent on the energy balance at the surface, flow speed, size and location, how the connection to geoengineering is made, I find lacking explanation (see comments below). My suggestion is to change the title to suit better the content of the paper.

The most interesting results and what I would think is the main results of this study, the differences between the different ESM are not really discussed and readers are left with more questions than answers. Looking at figures 4 and 5 there are many interesting things going on, but very little discussion and even misleading text, not presenting the results (for example line 145, see comment below). Why is there so big difference between the ESMs when the impact of the SAI is observed? Comparing the volume and area evolution for BNU-ESM and HadGEM2-ES it appears that the volume loss is reduced in the G4 simulations, but the reduction happens later in the BNU-ESM, the G4 line follows the RCP4.5 until about 2060, but the G4 line is off from RCP4.5 already in 2040 for HadGEM2-ES, why is this difference? For the MIROC runs the G4 lines (volume and area) follow the RCP4.5 lines. I think therefore that the numbers given in the abstract that G4 reduces mass loss from 16% to 12% misleading, as there is so big difference depending on which ESM is applied. The ensemble means and the numbers in the abstract are really showing the value in between the little MIROC response and the much larger HadGEM2-ES response to the SAI. Why are there such big differences in the responses? Also, very interesting is the area curves for the MIROC-ESM-CHEM results, the RCP8.5 reduces the area much slower than the RCP4.5 and G4 until about 2040 when it speeds up and overtakes in ca 2070 and the area and volume loss is larger than for the RCP4.5 and G4 runs. Similar, but smaller effect is also visible in the BNU-ESM results, the area (and volume) loss of RCP8.5 is slower in the first decades of the simulations but then speeds up and overtakes the RCP4.5 and G4 losses. The difference between the RCP4.5 and RCP8.5 volume and area loss is larger at 2089 in the MIROC runs than in the BNU-ESM and HadGEM2-ES, what causes this difference? I think the ensemble mean, shown in the figures furthest to the right is misleading and does

not give much information (as the numbers given in the abstract) what is interesting, and I find missing discussion of in the paper is the variable responses of the simulations forced with the different ESMs. There is no explanation of what impact G4 has on precipitation, temperature, or circulation in the model, that would be interesting, could this be added to the discussion?

The periods of the study are not consistently written through paper and it is confusing, in line 16 and 80 the period is stated 1982-2089, in line 61 2006-2089, line 96 period is 1982-1999 and in line 103 it is 1982-2005. In line 184 the 2089 is subtracted from 2020, is that present day reference (not 1999, or 2005/6?) My suggestion would be to have the periods, reference consistent through the paper.
Also, the period of the forcing is not consistent, in line 60 and 91 it is monthly, but in line 91 it is daily are both daily and monthly forcing used?

The description of the mass balance model is also not consistent and confusing, in line 79 SEMIC is introduced, but in line 82 it is stated that ESM is statistically downscaled and bias corrected using ISI-MIP, in line 97 it is stated that the spin-up is driven by SMB fields from PSIM forced with a sequence of ESM (no SEMIC or downscaling used?) in line 109 it is stated that SMB are corrected and SEMIC modelled and in line 117 it is stated that T, long wave and short wave radiation that drive SMB (SEMIC?) are bias-corrected (how?) with ERA5 reanalysis. My suggestion would be to straighten the description of what is done up and be consistent throughout the paper.

The whole section 4 reflects little or limited understanding of dynamics of ice caps and how the system responds to climate. See comments below. Ice cap in balance state loses mass at the edges and gains in the centre and the ice flow redistributes these to maintain the size and shape of equilibrated ice cap. The discussion in section 4 is strangely worded in many places and my suggestion would be to rewrite the whole section to better include known dynamics of ice caps and effect of SMB.

The Discussion section is confusing and has many unclear statements that don't make sense in the context of the presented study (see comments below) suggest reworking and clarifying and perhaps discussing the physical impacts of G4 on precipitation, temperature and why there is such a big difference between the 4 ESMs. In figure 8 results from 8 ESMs are presented, why are not all 8 used in the analysis before? The correlation between AMOC and SMB is shown, but there is no discussion of how this correlation might come about, there is no direct link, so some physical explanation of the relationship is missing.

Technical comments:

Abstract
Line 11-14 the first two sentence of the abstract are speculative and not useful as an entry for a paper that has title "Insensitivity of mass loss …." Suggest to state the findings of the study in the abstract to entice readers, not start with a speculative sentence: "SAI may reduce the mass loss by slowing surface temperature rise" does it, or does it not? (see comment above on title of the paper). The second sentence does not make sense: "although SMB is affected by the local climate, the sea level contribution is also dependent on ice dynamics" - this connection Although …. also … is strange, the sentence needs restructuring.
Line 17-19 this sentence is unclear, suggest to edit: "Ice dynamics are important for the ice cap loss rates … but making no difference to mass loss difference under the scenario"

Line 19-20 The following sentence does not make sense either and is not really supported by the material in the paper and conclusions: … "dynamics are remarkably insensitive to climate forcing " dynamics of ice caps are forced by geometry (slope, thickness) and rheology (ice viscosity) and therefore strange to relate to climate forcing or because "AMOC compensation to SMB and low rates of iceberg calving" suggest to rewrite this sentences. Also, the "AMOC compensation to SMB" is not shown in the paper and calving is not really discussed either, suggest to either delete or rewrite these statements. Line 21-22 this statement may be true, but is not supported by material in the paper, also the sentence reads strangely, suggest to edit and clarify and make a section in paper to support this statement.

Introduction
Line 26 "the unique climate" is strange here, every location on Earth really has unique climate, right? Suggest to edit sentence
Line 29 edit something strange here "which since"
Line 29 is there a reference supporting this statement?
Line 30 strange sentence, suggest to edit, glaciers in Iceland are very sensitive to changes in forcing and experience high mass throughput, Vatnajökull, the subject of this paper is however very large and is losing mass at slower rate than the neighboring Hofsjökull and Langjökull.
Line 31, suggest to delete "expected to accelerate" this is not shown in the references
Line 34 suggest to edit, strange sentence "obvious and deeply moving for Icelanders" what does that mean?
Line 37 more recent references, such as Aðalgeirsdóttir et al., 2020, Wouters et al., 2019 and Hugonnet et al., 2021 show that the mass loss rate has been slightly reduced after 2010 so this sentence should be edited.
Line 42-43 limiting global warming to less than 2°C is not an IPCC target, but the Paris agreement, IPCC is not prescriptive
Line 45 what does "relatively cheap way" mean here? Suggest to edit
Line 49 Vatnajökull is not in direct contact with the ocean (an outlet of Vatnajökull, Breiðamerkurjökull is calving into a lagoon that is connected with the ocean through a short river). Suggest to edit this sentence, calving and basal melt are not driven by changing climate or warming ocean
Line 50, suggest to delete "It is this component that we tackle here" see comment above
Line 51-53 this is very strange sentences, suggest to edit. The atypical behaviour of the North Atlantic is not discussed in this paper and neither is the compensatory effect of the climate forcing on the AMOC, suggest to either delete or explain better.
Line 55, is there a reference for this statement (warming at least twice as fast as the global mean)?
Line 57 missing "for" in front of "its"?
Line 57-58 not clear, what are "unwelcome impacts from geoengingeering"?
Lines 62-64, the descriptions of the two scenarios ("close to future emissions under the 2015 Paris agreement" and "extreme failure to mitigate scenario") are strange, suggest to use some other descriptor, like temperature by 2100 to describe these.

Line 66 Model and Verification, suggest to replace with "Validation", the convention is to use Verification for check if code is solving the equations right, but validate to compare to observations

Line 73, delete s in schemeS, suggest to replace "ice flow" with "constitutive equation"

Line 75, something is missing "Eigen sheme" does not make sense. Suggest to refer to PISM manual or website

Line 76 suggest to edit: "surface and bedrock elevation" or geometry, these two would provide the ice thickness, so it is redundant to include also ice thickness

Line 77, missing d in re-grided what does "these" mean here? From where are these data? Some reference to essential data for this study is missing. I would suggest to refer to Björnsson and Pálssson, 2020 for the bedrock data : https://www.cambridge.org/core/journals/annals-of-glaciology/article/radioecho-soundings-on-icelandic-temperate-glaciers-history-of-techniques-and-findings/4B1BDA5F075411D018245B4CEB7E9730) and surface mass balane a reference to Finnur Pálsson (2017) and maybe Aðalgeirsdóttir et al., where all smb data in Iceland is summariesed.

Line 78, see comment above, is the daily SMB filed used or monthly as stated in line 60?

Line 82-82, what does "lapse rate approach" mean? Do you correct with a temperature lapse rate? What is the value for the rate?

Line 83, what does "in reasonable agreement" mean? Some quantification or comparison would be useful here.

Line 86 (figure 1 caption) A) is not a location map, it only shows the Vatnajökull ice cap not where it is located in Iceland, suggest to put inset map that shows whole of Iceland and where Vatnajökull is located in figure 1a), not that one ' is missing in Tungnaárjökull (the second a should be á) , in d) is is the "annual average"? suggest to clarify

Line line 89 "equilibrium line boundary" is a strange wording, suggest to use the commonly used "equilibrium line altitude" , add something like "applied" or "assumed" before upward geothermal heat flux

Figure 1, see comment above, there is space in this figure (lower right corner) to add observed SMB that would aid the missing comparison with observation (see line 83)

Line 91, here it is stated that PISM is forced with monthly SMB fields (see comment line 78), what is the time resolution of the forcing?

Line 92-93 sentence is strange, something is missing, suggest something like: The final year of the spin-up simulation is then used as the initial condition in the experiments (or scenario simulations).

Line 96, figure 2 caption, suggest to add "simulation" after spin-up and also state if the forcing is annual, monthly or daily averaged over this period (hat is the time resolution of the forcing?) and also make sure the period is consistent, here it is stated 1982-1999, in Figure 1 the average surface mass balance is shown for the period 1982-2005.

Line 97 here it is stated that PISM is forced with 4 different ESM, is then the SEMIC model not used? See comment above, suggest to be consistent in describing the surface forcing method.

Line 99, it is strange to show the ensemble mean spatial distribution, as 2 of the models in the 4 piece ensemble have negative and 2 have positive difference, these could therefore cancel out in some location, suggest to either show only one, or all four, so it is possible to assess the performance of each simulation.

Line 101, is the magenta line the ensemble mean extent? See comment above, it is more useful to show each model separately.

Line 103 suggest to add a reference for SMB-altitude feedback.  Add "change" after elevation.  See comment above about the period, in caption for Figure 2 the period is stated 1982-1999

Line 104, suggest to use another word than "correct", It is not clear that the resulting SMB is more correct than the original (how can you assess that?), in equation it is called $SMB^{adj}$, why not call it then "adjusted" with more explanation?

Line 105 suggest to use different wording for "ESM-dependent "SMB lapse rate"" suggest to explain better what is meant and define what k is and how it is determined.

Line 109, see comment above, suggest to "adjusted" rather than "corrected"

Line 110, is this the modelled ice thickness in 2005? In Figure 2 is appears to be in year 1999 why is 2005 selected?  See comment above, how is k determined?

Line 112-113, this text reads awkwardly, suggest to use volume change for the evolution, but here write the difference between steady state and measured, or something like that.  Is the average over one year used? From Figure 2 it appears that the seasonal volume change is considerable.

Line 113  suggest to replace "Ice area loss" with difference between simulated state state and measured, see comment above. Suggest to replace "over" with "at"

Line 115, suggest to add "measured" before "ice thickness". Also suggest to use difference between steady state (or spin-up state) and measured, rather than "changes"

Line 116, this phrasing "are consistent across all the ESMs" is strange, suggest to write something like the spin-up steady states forced with the 4 ESM have similar steady-state geometry, or something like that

Line 117 here is strange wording, suggest to replace "that drive SMB" with something mentioning SEMIC model. Here is for first time the "bias-correction with ERA5 reanalysis mentioned, it should be clearer before that the all the ESM are "bias-corrected" with the same data. In line 82 it is stated that ESMs were bias corrected using ISI-MIP.  What does that actually mean? Are the annual or monthly averaged added or subtracted from the ESM values?

Line 118-124 this whole explanation is very confusing, suggest editing the whole paragraph.  The discrepancies are not caused by surging glaciers, the fact that most of the outlet glacier of Vatnajökull on the north and western side are surging and the model does not include any surging could be the reason for the model failing in simulating the observed ice thickness, that should be made clearer in this paragraph.  Suggest to take out "not parameterized" and use something like, not modelled or not included.

Line 127, In Table 1 only 2089 relative to 1982 is shown, not the difference duing 1991-2014, was that intended?

Line 127-128 neither the overestimation of SMB nor the disappearance of fast melting region are shown, more explanation is needed here.

Line 132, suggest to edit this sentence, it is very vague and more quantification and comparison would be useful,  "likely reason"  and "somewhat difference ice cap geometry" could be made clearer or better quantified.

Line 135-137 suggest to edit the whole figure caption and reconsider the ensemble and scenario averaged, suggest to show only one, or maybe two (there is space in the figure for at least, if not 3 more subfigures).  The text is redundant in two places "RCP4.5 and RCP8.5" are two times in same sentence and "average" and "mean", suggest to delete one of the two occurrences.

Line 137 suggest to replace "spaced" with "spatial resolution" and replace (upper left) with (upper right)

Line 139 see comment above Table 1 does not show historical changes as stated in lines 126 ad 140
Line

Line 141 are those 12% and 22% values relative to initial (which?) or maximum volume? It is not clear from text

Line 142, add "loss" after "volume"

Line 144 missing ' over second a in Tungnaárjökull

Line 145 This statement is not correct as shown in the 4th row of figure 5 for both the MIROC simulations, the difference is 0 (negative values are not shown, if there are any?) and the volume and area loss of G4 and RCP4.5 are very similar as shown in Figure 4

Line 145-146 this statement of G4 increasing ensemble ice thickness is strange, see comment above about ensemble mean not being useful, and that G4 increasing thickness is not true, the response of the model when G4 is that the thinning of the ice cap is reduced.

Line 149 see comment above, the ensemble mean is really not useful here, as it is taking the attention away from the interesting differences in the model responses.

Line 150 suggest to replace "Estimates considering ice dynamic from PISM" with "volume and area loss simulated by including ice dynamics"

Figure 5 in top line MIROC-ESM is misspelled as MIROE. The two bottom line figures should be shown with the same color scale for aiding comparison it is misleading to show differences with same color scale but different values, suggest to have both scales go to 100 m so that for example yellow color doesn't show 50 m in one and 70 m in the other row. It is not clear (figure caption states ice thickness differences between 2089 and 1982 is it the same initial state or ESM specific 1982 state? How different are the initial states at 1982?

Table 1 In this table no historical differences are shown as stated in Lines 127 and 139 (see comments above). See comments above that the ensemble mean with 4 ensemble members is not useful here. This table shows that very little difference is between the runs that couple ice dynamics with the SMB and the runs that have only SMB, therefore the statement in abstract line 18 seems an overestimate, how is ¼ and 1/3 difference found?

Line 163 "with maximum of more than 400 m" this seems large, given the mean thickness of the ice cap. Over how long period? What are the velocities that move this accumulated mass? Is this realistic or not?

Line 165-166 suggest to edite, "the smallest area of surface thinning" is strange wording. Also given the known higher temperature in RCP8.5 it is not surfacing that surface thinning is stronger for that scenario, by how much? Is even over the ice cap? Is it realistic differences? Why is there so little difference between RCP4.5 and RCP8.5 in the MIROC simulations?

Line 166 this sentence "Non-SMB components display the opposite pattern to SMB" should be deleted, it indicates little understanding of dynamics of ice cap.

Line 166-169 suggest to delete or edit this sentence to include ice dynamic understanding as it is written is seems like authors are analysing model results that are little understood.

Line 170-176 See comments above, the interesting results are that there is difference between the responses of the different ESM forcings, giving numbers for the ensemble (and showing in Figure 6) is hiding these interesting results.

Line 178-182 analysing the ensemble mean really hides the results shown in Figure 4, suggest to focus on that, rather than the ensemble mean with such small number of members and varying responses.

Figure 6 See comment above about the ensemble mean, the different responses between the 4 ESM is really interesting and that is lost in this figure that only shows the means and therefore misleading. Here the reference is year 2020 but both in Figure 5 and Table 1 the reference year is 1982, why not have the same reference in all figures and table? In figure b) large difference is between the dynamic (here called (dynamic), in (a) it is called (non-SMB), suggest to be consistent). How can the dynamic part be so different with same ice dynamic model? Figure 7 shows that the non-SMB part is very similar for all simulations, this figure is really strange showing such a large difference. The difference between G4 and RCP4.5 is very small, but Figure 4 shows that each of the ESM has very different response.

Line 192-196 see comment above, suggest to discuss separately each ESM response, as shown in Figure 7, than the mean. The large 95% confidence interval with N=4 clearly shows how variable the responses are.

Line 202-203 this sentence could be more clear, the non-SMB appears to have similar value throughout, which I think is clearer information than the the fraction becomes less important.

Line 207 this is strange, what about the impact on precipitation or temperature? I would think that it directly the forcing that impacts the response, rather than the degree of imbalance, could you confirm?

Line 209 "Iceland has been closer to balance until recently" is not very clear, what is recent here? The **glaciers** in Iceland were close to balance in period 1960-1995, after 1995 the mass balance became negative, and the rate of mass loss reduced after 2010.

Line 211 it is strange to discuss the relative effectiveness of SAI on reducing surface runoff, what is the effect on precipitation, temperature, atmospheric circulation?

Line 212 It is not clear what the "compensating impact of AMOC changes" are here, the correlaction between AMOC and SMB is shown, but what are the physical relationship? (what effect of precipitation and temperature are caused by AMOC changes?) this needs more discussion

Line 219 what is "SMB behavior" clarification is needed

Line 222 the sentence "may induce larger dynamic effects earlier" is not clear, needs editing. The dynamic effect appears to be very similar throughout the simulations as shown in Figure 7

Figure 8 Why are now 8 different ESM shown? Why are not all included in the analysis earlier in the paper?

Line 228 "annual mean maximum" is strange here, how is it both mean and maximum?

Line 236 "effects might be expected to be rather too small to be seen" is strange here, suggest to edit section and clarify

Line 239 something is missing "changing elevation-SMB" add "feedback"?

Line 242 not clear why "extreme maritime environment" (what is extreme about it?) makes a glacier most likely to exhibit a dynamical response, suggest to edit and clarify and also why such an effect I not seen in the experiment in this study.

Line 246 The sentence "Furthermore, retreat of the margins from the ocean" is not right here, there are no outlet glaciers of Vatnajökull residing in the ocean, the Jökulsárlón is inland lagoon, connected to the ocean by a river, but it is not ocean.

Line 251-251 sentence is strange and no connection between first and second part of it, suggest to edit.

Line 255 suggest to edite "in various basin ice thicknesses by 2089" does not make sense here

Line 258 what does "the relatively paramterized SEMIC model" mean, suggest to clarify

Line 259 suggest to edit "is still not perfectly captured" better to quantify, would you expect perfect capturing? When?

Line 260-261 strange sentence suggest to edit and clarify, not clear hoe albedo compensates for resolution?

Line 265 what is "de-weighting" suggest to edit

Line 265-268 strange sentences and suggest to edit, it is speculative "could perhaps provide improved polar impact studies

Lines 270-275 strange section and speculative, suggest to edit or delete

Line 270 what does "not particularly effective" mean?

Line 271 "unique geographical location" is strange, isn't every location unique? "we may infer" is strange here, suggest to delete

Line 272 sentence is strange "will not lead to greater mass loss of any glacier of ice cap" suggest to edit or delete

Line 274-275 suggest to delete. What is "palatable governance issues"? Moore et al., 2020 is not in reference list

Line 278 "reduces VIC mass loss by 4 percentage points" is strange, why not 4% ? suggest to edit

Line 279 "SAI could help preserve VIC from melting" is not true, the melting of the ice cap happens also in G4 simulations (suggest to replace "melt" with "mass loss" melting happens every summer)

Line 281 "compensating changes in temperature and accumulation due to AMOC" is not discussed before and should be better explained earlier in paper

Line 283 "VIC is relatively insensitive to climate scenario" does not make sense here ,suggest to edit or delete

Line 283 "relatively unaffected by changing air and ocean temperature" is not clear, ocean temperature does not affect dynamics as VIC is not in connection to ocean and the results of the study show that the dynamics is affected through changes in geometry of the ice cap. Suggest to edit or delete.

Line 384 the paper by Schmidt et al is now published and this reference should be replaced by the Cryosphere paper

Line 388 two places there should be ð instead of o: Aðalgeirsdóttir and Guðmundsson

---

## Referee Comment (RC2)

[referee-annotated manuscript omitted]

---

## Author Comment (AC1)

**Reviewer 1: Guðfinna Aðalgeirsdóttir**

**General comments**

This manuscript seems to me to be a follow up to a previous study by same authors published in Earth´s future (Yue et al. 2021) now with added ice flow model. The manuscript reads as uncompleted and hastily written afterthought that does not add much information to what was already published. Limited information about the models, limited understanding of ice dynamics (section 4 in particular) and poor presentation of the ensemble mean, rather than interesting results that the 4 ESM cause very different responses to the SAI, leaves reader with more questions than answers. Also, the fact that all the forcing fields are bias corrected (see comments below, some confusion about what is done) makes one wonder if any model dependent or physically caused impacts have been masked out with this bias correction and the observed responses therefore meaningless? Below are numerous comments about presentation and needs for clarifications. This manuscript needs major revisions.

Thanks, we have improved our manuscript significantly according to your valued suggestions. We have added the description of the four ESM responses to climate scenarios in Results section. The point of ISIMIP bias correction is to ensure the mean state of the model parameters match observations. The trend separate model trends over the observational period remain. This the bias correction ensures that models begin in close to an observed state. The separate ESM without bias correction have differences from observations e.g. several °C, and using these raw outputs would produce SMB that differed hugely from reality since those few degrees can make the difference between melting and not on the ice cap. The trends preserved by the bias correction allow very different future temperatures entirely driven by the ESM themselves. The commonly used method of looking at anomalies relative to a control scenario are not likely to work as well as bias correction where the non-linear change at the melting point in SMB mean that temperatures are important to get as correctly as possible. Thus, trend-preserving bias correction seems not only a logically consistent methodology, but an essential one if one wants to get an accurate SMB.

We have explained the statistically downscaling method more detailed. We have added a new section, 2.3: SMB modelling:
"In this study, the SMB fields used to drive PISM are from Yue et al. (2021), and estimated by SEMIC under the historical, G4, RCP4.5 and RCP8.5 scenarios during 1982–2089. SEMIC in turn is driven by downscaled and bias-corrected ESM data including temperatures, windspeeds, pressures, humidities and radiative forcing terms. We use all CMIP5 and GeoMIP ESM that have complete data fields available, namely BNU-ESM, HadGEM2-ES, MIROC-ESM, and MIROC-ESM-CHEM (Table 1). We statistically downscaled the ESM forcing based on the ERA5 reanalysis dataset (Hersbach et al., 2020). The point of the bias correction is to ensure the mean state of the model parameters matches observations. The separate model trends within each ESM over the observational period remain the same. Thus, the bias correction ensures that models begin close to an observed state, but can then diverge as the separate model climate dictate. The spatial resolution of ERA5 is about 30 km, but still cannot capture the VIC topography. To address this, we first downscaled ERA5 climate to 0.025°×0.025° grid based on their correlation with VIC surface elevation. We find surface elevation is well correlated with near-surface temperature (R=0.83, p<0.01), downward longwave (R=0.77, p<0.01) and shortwave radiation (R=0.74, p<0.01) and specific humidity (R=0.77, p<0.01), with lapse rates of -5.4 °C km$^{-1}$, -11.9 W m$^{-2}$ km$^{-1}$, 15.85 W m$^{-2}$ km$^{-1}$ and -0.59 k k$^{-1}$ km$^{-1}$, respectively. Precipitation and snowfall are downscaled following De Ruyter-de Wildt et al. (2004). The former is downscaled using Kriging interpolation method, with its empirically exponential relationship with observed surface elevation. The latter is assumed equal to precipitation rate when the daily mean air temperature is below 3°C, otherwise no snowfall occurs. Other SEMIC driven fields (surface wind speed, air density, pressure) are simply bilinearly interpolated due to the relatively minor effects on SMB in SEMIC. Then, we use the downscaled 0.025°×0.025° forcing fields as the observational reference climate to downscale and bias-correct the ESM fields using the ISIMIP approach (Hempel et al., 2013). The ISIMIP is a trend-preserving approach so that the long-term climate trends in models are preserved, while the mean at each grid cell is matched to observations. There are two fundamentally different ways ISIMIP can do the correction: addition and multiplication, and we follow ISIMIP protocol in deciding which method to use for each meteorological field variable (Hempel et al. 2013). The additive approach is used for most fields preserving, e.g. the absolute changes of the monthly temperature; while the multiplicative method is used for preserving the relative changes for precipitation and radiation. Finally, these 0.025°×0.025° fields were used to drive the SEMIC model. We also bias-corrected VIC surface albedo and considered SMB-elevation feedback in all simulations (Yue et al., 2021). Over the whole VIC, modelled SMB over the period 1991–2010 (Fig. 1d, Fig. 2) is well correlated (R=0.6, p<0.05) with an interpolated map from 60 measurement sites (Björnsson et al., 2013), although the mean is overestimated by 0.61 m yr$^{-1}$."

**Specific comments:**

The title of the manuscript is misleading and even misguiding. What is "solar geoengineering"? first guess would be that some engineering is done to the sun, this phrase is not used again in the paper, but "stratospheric aerosol injection" which is not directly related to "solar geoengineering", my suggestion is to be consistent throughout the paper about what is being discussed, injection in the stratosphere is not affecting the sun, is it? Also, the mass loss of ice caps is dependent on the energy balance at the surface, flow speed, size and location, how the connection to geoengineering is made, I find lacking explanation (see comments below). My suggestion is to change the title to suit better the content of the paper.

Solar geoengineering is the common umbrella terminology for technologies that alter shortwave radiative balance, and can be accomplished in many ways, but it seems to be unfamiliar. So based on your suggestion, we changed the title to "Insensitivity of mass loss of Icelandic Vatnajökull ice cap to stratospheric aerosol injection", and added the description about the "stratospheric aerosol injection" in Introduction section:

"Geoengineering by stratospheric aerosol injection (SAI) is designed to partially offset the longwave radiative forcing from increasing greenhouse gas concentrations in the atmosphere by reducing incoming solar radiation. Usually sulfate aerosols or their precursor, $SO_2$ are formulated in models, but other radiatively active aerosols have also been considered such as calcium carbonate or alumina (Angel, 2006, Cummings et al., 2017). The injection strategy may be global or designed to affect particular regions such as the Arctic (e.g. Robock et al., 2009), or designed to maintain particular useful constraints such as pole-equator temperature gradients (MacMartin and Kravitz, 2016)."

The most interesting results and what I would think is the main results of this study, the differences between the different ESM are not really discussed and readers are left with more questions than answers.

Looking at figures 4 and 5 there are many interesting things going on, but very little discussion and even misleading text, not presenting the results (for example line 145, see comment below). Why is there so big difference between the ESM when the impact of the SAI is observed? Comparing the volume and area evolution for BNU-ESM and HadGEM2-ES it appears that the volume loss is reduced in the G4

simulations, but the reduction happens later in the BNU-ESM, the G4 line follows the RCP4.5 until about

2060, but the G4 line is off from RCP4.5 already in 2040 for HadGEM2-ES, why is this difference?

The ice cap volume trend is largely determined by the SMB variability that forced PISM. We added an SMB figure in the main text, and a scatterplot between annual SMB and volume loss rate in the supplementary to show how temporal SMB changes in simulation scenarios that can explain the volume different behavior presented by ESM.

[Figure]

**Figure 2** Time series of annual (dotted curves) and decadal (solid curves) SMB during 1982–2089 under historical, G4 (red), RCP4.5 (blue) and RCP8.5 (black) modelled by SEMIC driven by downscaled and bias-corrected climate forcings from BNU-ESM, HadGEM2-ES, MIROC-ESM and MIROC-ESM-

CHEM, assuming a constant ice area for all simulations.

[Figure]

**Figure S1.** Scatterplot of annual SMB and volume loss rate over Vatnajökull ice cap under G4 (red), RCP4.5 (blue) and RCP8.5 (black) during 1982–2089 by BNU-ESM, HadGEM2-ES, MIROC-ESM and MIROC-ESM-CHEM.

Why is there so big difference between the ESM when the impact of the SAI is observed? Comparing the volume and area evolution for BNU-ESM and HadGEM2-ES it appears that the volume loss is reduced in the G4 simulations, but the reduction happens later in the BNU-ESM, the G4 line follows the RCP4.5 until about 2060, but the G4 line is off from RCP4.5 already in 2040 for HadGEM2-ES, why is this difference?

There are relatively small differences in SMB between RCP4.5 and G4. Each model has a single realization of each scenario. Therefore, differences between scenarios become noticeable by eye at different periods due to the variability of SMB and climate forcing over time. We added explanation in Section 3 to answer:

"Furthermore, there are small differences in the appearance of divergences between scenarios for each of the models, this is because there are random variations in weather and SMB forcing (Fig. 2). For example, the small differences in SMB between the G4 and RCP4.5 manifests itself in HadGEM2-ESM about 20 years earlier than BNU-ESM"

For the MIROC runs the G4 lines (volume and area) follow the RCP4.5 lines. I think therefore that the numbers given in the abstract that G4 reduces mass loss from 16% to 12% misleading, as there is so big difference depending on which ESM is applied. The ensemble means and the numbers in the abstract are really showing the value in between the little MIROC response and the much larger HadGEM2-ES response to the SAI. Why are there such big differences in the responses?

We revised abstract as your suggestion. The differences are due to the SMB forcing differences between models (see earlier plots), with MIROC differences between RCP4.5 and G4 being very small. We changed to:

"By 2089, G4 reduces VIC mass loss from 16 % under RCP4.5 to 12 % though with relatively large across-ESM spread. The SAI mitigating impacts are largely determined by SMB, with BNU-ESM and HadGEM2-ES having much larger changes than the two MIROC models."

Also, very interesting is the area curves for the MIROC-ESM-CHEM results, the RCP8.5 reduces the area much slower than the RCP4.5 and G4 until about 2040 when it speeds up and overtakes in ca 2070 and the area and volume loss is larger than for the RCP4.5 and G4 runs. Similar, but smaller effect is also visible in the BNU-ESM results, the area (and volume) loss of RCP8.5 is slower in the first decades of the simulations but then speeds up and overtakes the RCP4.5 and G4 losses. The difference between the RCP4.5 and RCP8.5 volume and area loss is larger at 2089 in the MIROC runs than in the BNU-ESM and HadGEM2-ES, what causes this difference?

We answer your question in the Section 3, we added:

"Area loss rates under RCP8.5 are smaller than RCP4.5 and G4 prior 2040 with BNU-ESM and MIROC-ESM-CHEM, but later, loss rates under RCP8.5 accelerate eventually having larger area loss than RCP4.5 after 2080 for BNU-ESM and after 2075 for MIROC-ESM-CHEM. The main reason is again due to SMB, and is fundamentally due by the slightly lower VIC near-surface air temperature under RCP8.5 before 2035. Despite RCP8.5 being a high emissions scenario, the differences in radiative forcing between scenarios are smaller than random climate variability in the first few decades of the 21$^{st}$ century. Beyond the 2050s, the higher temperatures, surface downward longwave radiation fluxes as well as lower snowfall in RCP8.5 (Yue et al., 2021) become more significantly different from other scenarios. By 2089, the volume and area differences between RCP4.5 and RCP8.5 are larger in the MIROC runs than in the BNU-ESM and HadGEM2-ES. This is clearly due to mean SMB differences (RCP8.5-RCP4.5) during 2006–2089: -0.20, -0.25, -0.42, -0.40 m yr$^{-1}$ for BNU-ESM, HadGEM2-ES, MIROC-ESM and MIROC-ESM-CHEM, respectively (Fig. 2)."

I think the ensemble mean, shown in the figures furthest to the right is misleading and does not give much information (as the numbers given in the abstract) what is interesting, and I find missing discussion of in the paper is the variable responses of the simulations forced with the different ESM.

We wanted to avoid talking about stochastic variability "weather" rather than actual significant differences due to scenario. There is always large across-model spread for ESM. This is why the ensemble mean is so popular, e.g. in IPCC reports. However, we revised Section 3, to describe more results presented for individual ESM, avoiding the misleading by ensemble mean. We define uncertainties in this study as the ensemble mean and 95% confidence interval, N=4. We added:

"G4 reduces the VIC volume and area by 4±4 % and 2±3 % relative to RCP4.5. The relatively large spread demonstrates the different SAI impacts across each ESM, e.g., G4 reduces the VIC volume 7–8 % relative to RCP4.5 with BNU-ESM and HadGEM2-ES forcing, but the two MIROC models predict little differences. These are mainly determined by SMB in these scenarios, G4 reduces SMB by 0.25 m yr$^{-1}$ and 0.41 m yr$^{-1}$ during 2020–2069 in BNU-ESM and HadGEM2-ES, but less than 0.11 m yr$^{-1}$ in the two MIROC models (Fig. 2)."

There is no explanation of what impact G4 has on precipitation, temperature, or circulation in the model, that would be interesting, could this be added to the discussion?

Done. We added:

"In G4, changes in Atlantic Ocean circulation may increase VIC temperatures. Projections by all ESM with data show AMOC index at 30°N is 0–4 Sv stronger in G4 than RCP4.5 (Fig. 9a), which acts to increase heat flux from ocean to atmosphere near Iceland (Fig. 9d). However, the atmospheric cooling associated with G4 SAI dominates the VIC climate, resulting in a 0.4°C reduction of air temperature and a 6% lower surface melt-runoff under G4. There are across model differences, with the two MIROC projecting little changes between G4 and RCP4.5 in temperatures and precipitation, and hence the response of ice cap volume. Precipitation is the main component of mass accumulation, all ESM project insignificant precipitation differences between G4 and RCP4.5. This is different from the global (Trisos et al., 2018) and Greenland (Moore et al., 2019) cases where G4 reduces precipitation in most regions, due to the fundamental difference between long wave greenhouse gas and shortwave SAI radiative forcing. Greenhouse gases are distributed throughout the atmosphere, while shortwave radiation impacts surface temperatures, hence temperature lapse rates are altered under SAI and the atmosphere is drier than it would be for the same temperature under simple greenhouse gas climates. The changes precipitation under G4 that are seen in VIC may be driven by the relatively enhanced AMOC and lower Arctic sea ice (Xie et al., 2022) which in turn brings more water vapor to VIC."
"

The periods of the study are not consistently written through paper and it is confusing, in line 16 and 80 the period is stated 1982-2089, in line 61 2006-2089, line 96 period is 1982-1999 and in line 103 it is 1982-2005. In line 184 the 2089 is subtracted from 2020, is that present day reference (not 1999, or 2005/6?)   My suggestion would be to have the periods, reference consistent through the paper.

Sorry, we can't use a common reference period, it's better to follow the CMIP5 scenario period definition that 2005 is the line between the historical and future. As for 1982–1999 which is the spin-up period followed by Schmidt et al. (2020) because VIC was close to steady state during that period. G4 is designed in 2020-2089, but the aerosol injection only over 2020-2069, so, we compare both periods. We revised descriptions in Section Introduction to make it clearer:

"We simulate the response of the VIC with the Parallel Ice Sheet Model (PISM; version 1.0) driven by monthly SMB from 1982–2089 under CMIP5 historical (1982–2005), RCP4.5 (2006–2089), RCP8.5 (2006–2089) and SAI G4 (2020–2089) scenarios. The SMB fields are modelled by a surface energy and mass balance model (Section 2.1 and 2.3) driven by downscaled and bias-corrected climate forcings by all Earth System Model (ESM; Table 1) that have sufficient data fields available from both RCP and G4 scenarios. RCP4.5 (Thomson et al., 2011) is a stabilization scenario with emissions similar to those agreed under the Paris 2015 agreement (Kitous and Keramidas, 2015), while RCP8.5 (Riahi et al., 2011) is a "business-as-usual" scenario that is a likely outcome if we do not make any efforts to reduce the greenhouse gas emissions. By the end of the 21$^{st}$ century, their total radiative forcing is stabilized at roughly 4.5 and 8.5 W m$^{-2}$, and with global mean surface temperature rise by 1.8 and 3.7 °C relative to 1986–2005 (IPCC, 2014). The SAI G4 scenario branches off the RCP4.5 scenario in 2020, specifying 5 Tg yr$^{-1}$ of SO$_2$ to be injected into the equatorial lower stratosphere until 2069, and then continues with RCP4.5 forcing to 2089 (Kravitz et al., 2013). We quantitively evaluate the SAI G4 impact by analyzing differences of the VIC geometry between 2020 and 2069, as well as the whole simulation period between 1982 and 2089."

Also, the period of the forcing is not consistent, in line 60 and 91 it is monthly, but in line 91 it is daily are both daily and monthly forcing used?

No, PISM is only driven by monthly SMB, which are from daily SMB modelled by SEMIC. We corrected the error:

"To initialize PISM over the VIC, we need the boundary conditions of the surface elevation, bedrock
altitude, upward geothermal flux, ice temperature, and monthly surface mass balance (Table 1, Fig. 1,
Fig. 2)."

The description of the mass balance model is also not consistent and confusing, in line 79 SEMIC is
introduced, but in line 82 it is stated that ESM is statistically downscaled and bias corrected using ISI-
MIP, in line 97 it is stated that the spin-up is driven by SMB fields from PSIM forced with a sequence of
ESM (no SEMIC or downscaling used?) in line 109 it is stated that SMB are corrected and SEMIC
modelled and in line 117 it is stated that T, long wave and short wave radiation that drive SMB (SEMIC?)
are bias-corrected (how?) with ERA5 reanalysis. My suggestion would be to straighten the description
of what is done up and be consistent throughout the paper.
Climate fields from ESM are downscaled and bias-corrected to 0.025°, and then these fields are used to
calculate SMB by SEMIC, so the SMB resolution is also 0.025°, we use the 0.025° SMB to run PISM.
We added a Section 2.3 'SMB modelling' to describe how we downscale the ESM output and how we
estimate SMB by SEMIC model:
"In this study, the SMB fields used to drive PISM are from Yue et al. (2021), and estimated by SEMIC
under the historical, G4, RCP4.5 and RCP8.5 scenarios during 1982–2089. SEMIC in turn is driven by
downscaled and bias-corrected ESM data including temperatures, windspeeds, pressures, humidities and
radiative forcing terms. We use all CMIP5 and GeoMIP ESM that have complete data fields available,
namely BNU-ESM, HadGEM2-ES, MIROC-ESM, and MIROC-ESM-CHEM (Table 1). We statistically
downscaled the ESM forcing based on the ERA5 reanalysis dataset (Hersbach et al., 2020). The point of
the bias correction is to ensure the mean state of the model parameters matches observations. The separate
model trends within each ESM over the observational period remain the same. Thus, the bias correction
ensures that models begin close to an observed state, but can then diverge as the separate model climate
dictate. The spatial resolution of ERA5 is about 30 km, but still cannot capture the VIC topography. To
address this, we first downscaled ERA5 climate to 0.025°×0.025° grid based on their correlation with
VIC surface elevation. We find surface elevation is well correlated with near-surface temperature
(R=0.83, p<0.01), downward longwave (R=0.77, p<0.01) and shortwave radiation (R=0.74, p<0.01) and
specific humidity (R=0.77, p<0.01), with lapse rates of -5.4 °C km$^{-1}$, -11.9 W m$^{-2}$ km$^{-1}$, 15.85 W m$^{-2}$ km$^{-1}$
$^{1}$ and -0.59 k k$^{-1}$ km$^{-1}$, respectively. Precipitation and snowfall are downscaled following De Ruyter-de
Wildt et al. (2004). The former is downscaled using Kriging interpolation method, with its empirically
exponential relationship with observed surface elevation. The latter is assumed equal to precipitation rate
when the daily mean air temperature is below 3°C, otherwise no snowfall occurs. Other SEMIC driven
fields (surface wind speed, air density, pressure) are simply bilinearly interpolated due to the relatively
minor effects on SMB in SEMIC. Then, we use the downscaled 0.025°×0.025° forcing fields as the
observational reference climate to downscale and bias-correct the ESM fields using the ISIMIP approach
(Hempel et al., 2013). The ISIMIP is a trend-preserving approach so that the long-term climate trends in
models are preserved, while the mean at each grid cell is matched to observations. There are two
fundamentally different ways ISIMIP can do the correction: addition and multiplication, and we follow
ISIMIP protocol in deciding which method to use for each meteorological field variable (Hempel et al.
2013). The additive approach is used for most fields preserving, e.g. the absolute changes of the monthly
temperature; while the multiplicative method is used for preserving the relative changes for precipitation
and radiation. Finally, these 0.025°×0.025° fields were used to drive the SEMIC model. We also bias-
corrected VIC surface albedo and considered SMB-elevation feedback in all simulations (Yue et al.,

2021). Over the whole VIC, modelled SMB over the period 1991–2010 (Fig. 1d, Fig. 2) is well correlated
(R=0.6, p<0.05) with an interpolated map from 60 measurement sites (Björnsson et al., 2013), although
the mean is overestimated by 0.61 m $yr^{-1}$."

The whole section 4 reflects little or limited understanding of dynamics of ice caps and how the system
responds to climate. See comments below. Ice cap in balance state loses mass at the edges and gains in
the centre and the ice flow redistributes these to maintain the size and shape of equilibrated ice cap.   The
discussion in section 4 is strangely worded in many places and my suggestion would be to rewrite the
whole section to better include known dynamics of ice caps and effect of SMB.
We revised the Section 4. See revisions below your every comment. We naturally disagree that we have
little understanding of ice dynamics, and instead suggest that the difficulties were with inadequate
explanations. The authors include experienced ice dynamics modelers with a proven track record
published research, for example modelling VIC with PISM (Schmidt, et al   2020 *J. Glaciol*.,
doi:10.1017/jog.2019.90); using higher order models in Greenland drainage basins (Guo, et al 2019, *The*
*Cryosphere,* https://doi.org/10.5194/tc-13-3139-2019); using full Stokes model for Antarctica ice domes
(Zhao, et al. 2018 *The Cryosphere,* doi:10.5194/tc-12-1651-2018) and small glaciers in Asia (Zhao, et al.
2013 *J. Glaciology,* doi: 10.3189/2014JoG13J126; Zhao, et al. 2022, *Water*,
https://doi.org/10.3390/w14020271); developing and using combined ice dynamic and basal hydrology
models (Wolovick, et al 2021a *JGR* https://doi.org/10.1029/2020JF005937; Wolovick, et al, 202b1 *JGR*
https://doi.org/10.1029/2020JF005936).

The Discussion section is confusing and has many unclear statements that don't make sense in the context
of the presented study (see comments below) suggest reworking and clarifying and perhaps discussing
the physical impacts of G4 on precipitation, temperature and why there is such a big difference between
the 4 ESM.
Sorry about that, we have endeavored to address specific comments and looked at the section again. We
added impacts of G4 on precipitation and temperature:
"In G4, changes in Atlantic Ocean circulation may increase VIC temperatures. Projections by all ESM
with data show AMOC index at 30°N is 0–4 Sv stronger in G4 than RCP4.5 (Fig. 9a), which acts to
increase heat flux from ocean to atmosphere near Iceland (Fig. 9d). However, the atmospheric cooling
associated with G4 SAI dominates the VIC climate, resulting in a 0.4°C reduction of air temperature and
a 6 % lower surface melt-runoff under G4. There are across model differences, with the two MIROC
projecting little changes between G4 and RCP4.5 in temperatures and precipitation, and hence the
response of ice cap volume. Precipitation is the main component of mass accumulation, all ESM project
insignificant precipitation differences between G4 and RCP4.5. This is different from the global (Trisos
et al., 2018) and Greenland (Moore et al., 2019) cases where G4 reduces precipitation in most regions,
due to the fundamental difference between long wave greenhouse gas and shortwave SAI radiative
forcing. Greenhouse gases are distributed throughout the atmosphere, while short wave radiation impacts
surface temperatures, hence temperature lapse rates are altered under SAI and the atmosphere is drier
than it would be for the same temperature under simple greenhouse gas climates. The changes
precipitation under G4 that are seen in VIC may be driven by the relatively enhanced AMOC and lower
Arctic sea ice (Xie et al., 2022) which in turn brings more water vapor to VIC."
Regarding differences between ESM – these 4 ESM are within the typical range of equilibrium climate
sensitivity (i.e. the global mean surface air temperature change caused by a doubling of the atmospheric

$CO_2$ with the BNU-ESM, HadGEM2-ES and MIROC models {3.92, 4.61, 4.67} K) exhibited by CMIP5

models. There is a range of climate responses, and when small regions such as Iceland are the focus, the differences between ESM are naturally larger than when averaged over the globe or larger regions, simply by the central limit theorem.

In figure 8 results from 8 ESM are presented, why are not all 8 used in the analysis before? The correlation between AMOC and SMB is shown, but there is no discussion of how this correlation might come about, there is no direct link, so some physical explanation of the relationship is missing.

Because for SMB modelling, SEMIC needs 8 daily climate fields to estimate SMB, and there are only 4

ESM with all the data available in both G4 and RCP scenarios. We use every possible model. For the

AMOC, we use all 8 ESM that have done G4, and which is consistent with the GrIS mass balance data from Goelzer et al. (2021). In Section 2.3 SMB modelling, we added:

"In this study, the SMB fields used to drive PISM are from Yue et al. (2021), and estimated by SEMIC

under the historical, G4, RCP4.5 and RCP8.5 scenarios during 1982–2089. SEMIC in turn is driven by downscaled and bias-corrected ESM data including temperatures, windspeeds, pressures, humidities and radiative forcing terms. We use all CMIP5 and GeoMIP ESM that have complete data fields available, namely BNU-ESM, HadGEM2-ES, MIROC-ESM, and MIROC-ESM-CHEM (Table 1)."

**Technical comments**

**Abstract**

Line 11-14 the first two sentence of the abstract are speculative and not useful as an entry for a paper that has title "Insensitivity of mass loss …." Suggest to state the findings of the study in the abstract to entice readers, not start with a speculative sentence: "SAI may reduce the mass loss by slowing surface temperature rise" does it, or does it not? (see comment above on title of the paper). The second sentence does not make sense: "although SMB is affected by the local climate, the sea level contribution is also dependent on ice dynamics" – this connection Although …. Also … is strange, the sentence needs restructuring.

We rewrote the abstract as follows:

**Abstract.** Geoengineering by stratospheric aerosol injection (SAI) impacts the North Atlantic region differently from the rest of the world, because in climate models it reverses the slow-down in the Atlantic Meridional Circulation (AMOC) driven by greenhouse gas warming. AMOC delivers significant heat to Iceland, and hence plays an important role in determining mass loss from the Vatnajökull ice cap (VIC). We use the Parallel Ice Sheet Model (PISM) to estimate the VIC mass balance under the CMIP5 (Coupled Model Intercomparison Project Phase 5) RCP4.5, RCP8.5 and GeoMIP (Geoengineering Model Intercomparison Project) G4 SAI scenarios during the period 1982–2089, driven by statistically downscaled climate forcings from four Earth System Models (ESM). The G4 scenario follows the greenhouse gas emissions trajectory specified by RCP4.5, but with additional 5 Tg $yr^{-1}$ of $SO_2$ injection to the lower stratosphere. By 2089, G4 reduces VIC mass loss from 16 % under RCP4.5 to 12 % though with relatively large across-ESM spread. The SAI mitigating impacts are largely determined by SMB, with BNU-ESM and HadGEM2-ES having much larger changes than the two MIROC models. All ESM show that the non-SMB component (i.e., ice dynamics and basal melting) remains nearly constant at around -0.25 m $yr^{-1}$ and is remarkably insensitive to climate forcing over time for all scenarios. This non-SMB component is important for ice cap loss rates compared with mass balances of -0.47, -0.61 and -0.88 m $yr^{-1}$ over the 1982–2089 period under G4, RCP4.5 and RCP8.5, respectively. The unusually stable dynamic losses are consistent with the much higher geothermal heat flows under parts of the ice cap than in most glaciers elsewhere.

Line 17-19 this sentence is unclear, suggest to edit: "Ice dynamics are important for the ice cap loss rates … but making no difference to mass loss difference under the scenario"

We corrected:

"All ESM show that the non-SMB component (i.e., ice dynamics and basal melting) remain nearly constant at around -0.25 m $yr^{-1}$, and is remarkably insensitive to climate forcing over time for all scenarios. This non-SMB component is important for ice cap loss rates compared with mass balances of -0.47, -0.61 and -0.88 m $yr^{-1}$ over the 1982–2089 under G4, RCP4.5 and RCP8.5, respectively."

Line 19-20 The following sentence does not make sense either and is not really supported by the material in the paper and conclusions: … "dynamics are remarkably insensitive to climate forcing "dynamics of ice caps are forced by geometry (slope, thickness) and rheology (ice viscosity) and therefore strange to relate to climate forcing

We disagree, it is not strange that climate forcing affects ice dynamics. Climate forcing affects ice dynamics in several ways, of relevance here is it increases ablation around the edge of the ice cap, in most glaciers high altitude snowfall either is pretty constant or increases in greenhouse gas scenarios, leading to steeping of the ice. In the case of the Greenland and Antarctic ice sheets, grounding line retreat leads to dynamic changes that depends on ocean thermal forcing that depends on climate scenario. On longer timescales climate warming likely warms the ice, in the case of cold glaciers, or changes the quantity of water within the ice, both of which changes its viscosity.

"All ESM show that the non-SMB component (i.e., ice dynamics and basal melting) remain nearly constant at around -0.25 m yr$^{-1}$, and is remarkably insensitive to climate forcing over time for all scenarios."

Or because "AMOC compensation to SMB and low rates of iceberg calving" suggest to rewrite this sentences. Also, the "AMOC compensation to SMB" is not shown in the paper and calving is not really discussed either, suggest to either delete or rewrite these statements.

Ok, we deleted this sentence.

Line 21-22 this statement may be true, but is not supported by material in the paper, also the sentence reads strangely, suggest to edit and clarify and make a section in paper to support this statement.

OK, we deleted this sentence.

**1 Introduction**

Line 26   "the unique climate" is strange here, every location on Earth really has unique climate, right?

Suggest to edit sentence

Deleted "unique", and replaced with: "the unusual and particular climate of Iceland"

Line 29 edit something strange here "which since"

We deleted which.

Line 29 is there a reference supporting this statement?

Added: Oerlemans, J. (1992). Climate sensitivity of glaciers in southern Norway: Application of an energy-balance model to Nigardsbreen, Hellstugubreen and Alfotbreen. *Journal of Glaciology, 38*(129),

223-232. Doi:10.3189/S0022143000003634; Rupper, S., & Roe, G. (2008). Glacier Changes and

Regional Climate: A Mass and Energy Balance Approach, *Journal of Climate*, *21*(20), 5384-5401

Line 30 strange sentence, suggest to edit, glaciers in Iceland are very sensitive to changes in forcing and experience high mass throughput, Vatnajökull, the subject of this paper is however very large and is losing mass at slower rate than the neighboring Hofsjökull and Langjökull

We removed our previous sentence, and followed your edit:

"Glaciers in Iceland are very sensitive to changes in forcing and experience high mass throughput, since maritime glaciers are more sensitive to climate variations than continental ones (Oerlemans, 1992).

Vatnajökull ice cap, the subject of this paper, is however very large and is losing mass at slower rate than the neighboring Hofsjökull and Langjökull ice cap (Björnsson et al., 2002, Jóhannesson et al., 2006)."

Line 31, suggest to delete "expected to accelerate" this is not shown in the references

Done.

Line 34 suggest to edit, strange sentence "obvious and deeply moving for Icelanders" what does that
mean?

It means that many Icelanders that we have spoken and worked with enjoy and identify with their land
having glaciers. In other parts of the world, loss of ice cover has had exactly the impact queried here e.g.
in artistic interpretation and emotional attachment to the landscape (Orlove, B., E. Wiegandt and B. H.
Luckman (eds.) 2008. Darkening Peaks. Glacier Retreat, Science, and Society. Berkeley and Los Angeles:
University of California Press). That might be expected of Icelanders as well. But since this impact is not
particularly relevant here we delete that, and revised to:
"Although their contribution to global mean sea-level rise would be just 1 cm, even if all the ice melted
(Björnsson and Pálsson, 2008), the local impacts of rapid glacier loss will be obvious and will cause
profound changes in hydrology (Flowers et al., 2003)."

Line 37 more recent references, such as Aðalgeirsdóttir et al., 2020, Wouters et al., 2019 and Hugonnet
et al., 2021 show that the mass loss rate has been slightly reduced after 2010 so this sentence should be
edited.

Ok, done, but we do not sure which article Wouters et al., 2019 refers to, we revised to:
"Surface mass balance (SMB, the sum of accumulation and ablation) significantly decreased from a
slightly positive balance in the 1980s to -0.8 m yr$^{-1}$ during 1995-2014 (Pálsson et al., 2017), but mass
loss rate slightly reduced after 2010 (Aðalgeirsdóttir et al., 2020, Hugonnet et al., 2021)."

Line 42-43 limiting global warming to less than 2°C is not an IPCC target, but the Paris agreement, IPCC
is not prescriptive

Ok, done.

Line 45 what does "relatively cheap way" mean here? Suggest to edit

It's the financially cost for the implementation of SAI. We revised to:
"Moreover, deployment of SAI may be a financially cheap way to offset temperature rises on the global
scale (Smith and Wagner, 2018)."

Line 49 Vatnajökull is not in direct contact with the ocean (an outlet of Vatnajökull, Breiðamerkurjökull
is calving into a lagoon that is connected with the ocean through a short river). Suggest to edit this
sentence, calving and basal melt are not driven by changing climate or warming ocean

Thanks for your explanation. We deleted this sentence "that are driven by changing climate or impacts
due to the warming ocean in contact with the ice" and revised to:
"However, Yue et al. (2021) did not consider non-surface mass balance generated by changes in ice flow
and discharge (e.g., calving of ice and basal melting)."

Line 50, suggest to delete "It is this component that we tackle here" see comment above

Done.

Line 51-53 this is very strange sentences, suggest to edit. The atypical behaviour of the North Atlantic is
not discussed in this paper and neither is the compensatory effect of the climate forcing on the AMOC,
suggest to either delete or explain better.

Rewritten as :
"here we focus only on impacts from SAI on the mass balance of a single ice cap in Iceland. The topic
is of wider interest because the behaviour of the North Atlantic under both climate models driven by
greenhouse gases, and observational evidence points to a slow-down in AMOC, leading to a much-
reduced rate of warming in the North Atlantic relative to the rest of world (Cheng et al., 2013). Under
SAI, AMOC slows less than under greenhouse gas climates (Hong et al., 2017; Yue et al., 2021; Xie et
al., 2022). Thus, in Iceland, we would expect SAI changes on AMOC and radiative forcing to have
compensatory effects to the ice cap. Furthermore, the Arctic warmed 6 times faster than the global mean
from 1998-2012 (Huang et al., 2017), leading to concerns on the stability of the Arctic cryosphere, and
examination of possible roles for geoengineering methods in its preservation (Lee et al., 2021).Whether
SAI might even lead to exacerbated ice mass loss in the North Atlantic is an important question that goes
to the fundamental reason for ever doing SAI – that is does SAI better preserve the important elements
of the current climate system than plausible greenhouse gas emissions scenarios?"

Line 55, is there a reference for this statement (warming at least twice as fast as the global mean)?
Done.
"Furthermore, the Arctic is warmed 6 times faster than the global mean from 1998-2012, (Huang, J. et
al. Recently amplified arctic warming has contributed to a continual global warming trend. Nat. Clim.
Chang. 7, 875–879 (2017)."

Line 57 missing "for" in front of "its"?
Inserted "in", not for.

Line 57-58 not clear, what are "unwelcome impacts from geoengineering"?
Unwelcome impacts mean the geoengineering may fail in this region due to the Arctic amplification and
the impact of enhanced AMOC under geoengineering. See reply of Line 51-53.

Lines 62-64, the descriptions of the two scenarios ("close to future emissions under the 2015 Paris
agreement" and "extreme failure to mitigate scenario") are strange, suggest to use some other descriptor,
like temperature by 2100 to describe these.
It is important from the policy relevance perspective that RCP4.5 is close to the Paris 2015 agreement.
But we edited it:
"RCP4.5 (Thomson et al., 2011) is a stabilization scenario with emissions similar to those agreed under
the Paris 2015 agreement (Kitous and Keramidas, 2015), while RCP8.5 (Riahi et al., 2011) is a "business-
as-usual" scenario that is a likely outcome if we do not make any efforts to reduce the greenhouse gas
emissions. By the end of the 21$^{st}$ century, their total radiative forcing is stabilized at roughly 4.5 and 8.5
W m$^{-2}$, and with global mean surface temperature rise by 1.8 and 3.7 °C relative to 1986–2005 (IPCC,
2014)."

**2 Model and validation**

Line 66 Model and Verification, suggest to replace with "Validation", the convention is to use Verification for check if code is solving the equations right, but validate to compare to observations

Yes, thanks. Done

Line 73, delete s in schemeS, suggest to replace "ice flow" with "constitutive equation"

Done.

Line 75, something is missing "Eigen scheme" does not make sense. Suggest to refer to PISM manual or website

We gave a description for "Eigen scheme". We added:

"—Ice front calving rate c is calculated by the strain rate Eigenvalue scheme (Levermann et al., 2012):

$$c = K \cdot \max(0, \epsilon_\parallel) \cdot \max(0, \epsilon_\perp) \tag{5}$$

Where K is a constant that explains the ice properties relevant to calving, $\epsilon_\parallel$ and $\epsilon_\perp$ denote the strain rate along and transversal to horizontal ice flow, respectively.

We also added some brief descriptions about PISM model and parameterizations we used:

The PISM model (version 1.0; Bueler and Brown (2009); https://www.pism.io) is an open-source ice sheet thermo-dynamic model that has been used in numerous studies of a wide range of ice sheets and glaciers (e.g., Aschwanden et al., 2019; Yan et al., 2020). The evolution of the ice cap surface elevation *H* is calculated by mass continuity equation:

$$\frac{dH}{dt} = M - \nabla \cdot \vec{Q} - M_b \tag{2}$$

Where *t* is the time step, *M* is the mass balance, $M_b$ is the basal melt rate, $\nabla \cdot \vec{Q}$ is the ice flux calculated by stress balance model. PISM model provides several parameterizations to describe the ice stress balance, ice flow, basal sliding and ice calving (details see PISM manual; https://www.pism.io/docs/). The choices of parameterizations and free parameters followed Schmidt et al. (2020), and validated the simulations using observations over Vatnajökull. In brief the parameterizations we used in this study are::

—We use hybrid stress balance model (Bueler and Brown, 2009) with both Shallow Ice Approximation (SIA; Hutter, 1983) and Shallow Shelf Approximation (SSA; Morland, 1987) to solve ice vertical deformation and longitudinal stretching, allowing simulation of both slowly flowing ice cap interiors and fast flowing outlet glaciers.

—Ice rheology is parameterized by Glen's flow law (Glen, 1955):

$$\tau = 2\eta D \,, \tag{3}$$

where $\tau$ is the deviatoric stress tensor, *D* is the strain rate tensor, and $\eta$ is given by:

$$\eta = \frac{1}{2} A(T)^{-1/n} d_e^{(1-n)/n}, \tag{4}$$

where the parameter *A* is strongly dependent on ice temperature, $d_e$ is the second invariant of the strain rate tensor, flow exponent *n* is commonly taken the value of 3.

—Ice front calving rate $c$ is calculated by the strain rate Eigenvalue scheme (Levermann et al., 2012):

$$c = K \cdot \max(0, \epsilon_\parallel) \cdot \max(0, \epsilon_\perp) \tag{5}$$

Where $K$ is a constant that explains the ice properties relevant to calving, $\epsilon_\parallel$ and $\epsilon_\perp$ denote the strain rate along and transversal to horizontal ice flow, respectively.

—Basal sliding is estimated by pseudo-plastic law (Bueler and Brown, 2009), which estimate the basal shear stress $\tau_b$ through the yield stress $\tau_c$, basal velocity $u_b$, and parameters of velocity threshold $u_{threshold}$ and power $q$:

$$\tau_b = -\tau_c \frac{u_b}{u_{threshold}^q |u_b|^{1-q}} \tag{6}$$

Line 76 suggest to edit: "surface and bedrock elevation" or geometry, these two would provide the ice thickness, so it is redundant to include also ice thickness

We corrected:

"To initialize PISM over the VIC, we need surface elevation, bedrock altitude, upward geothermal flux, ice temperature, and monthly surface mass balance (Table 1, Fig. 1, Fig. 2)."

Line 77, missing d in re-grided   what does "these" mean here? From where are these data? Some reference to essential data for this study is missing. I would suggest to refer to Björnsson and Pálsson, 2020 for the bedrock data : https://www.cambridge.org/core/journals/annals-of-glaciology/article/radioecho-soundings-on-icelandic-temperate-glaciers-history-of-techniques-and-findings/4B1BDA5F075411D018245B4CEB7E9730) and surface mass balane a reference to Finnur Pálsson (2017) and maybe Aðalgeirsdóttir et al., where all smb data in Iceland is summariesed.

The PISM input data is followed by Schmidt et al. (2020), we cited the bedrock data from Björnsson and Pálssson, 2020 and we made a table to describe these data:

**Table 1** A summary input data fields in PISM.

| PISM input fields | Data source | Period | PISM running resolution | Reference |
|---|---|---|---|---|
| Surface mass balance | SEMIC output driven by downscaled and bias-corrected climate fields from [a] BNU-ESM, [b] HadGEM2-ES, [c] MIROC-ESM, [d] MIROC-ESM-CHEM | 1982–1999, repeated for 2000 years (PISM spin-up) 1982–2005 (CMIP5 historical) 2006–2089 (RCP4.5, RCP8.5) 2020–2089 (GeoMIP G4) | Monthly; 500 ×500 m | Yue et al. (2021) |
| Surface elevation | Spot5 satellite | June to September 2010 | 500 ×500 m | Berthier and Toutin. (2008) |
| Bedrock topography | Radio echo profiles | 1980 | 500 ×500 m | Björnsson, (1986); Björnsson and Pálsson. (2020) |
| Ice cap thickness | Surface elevation minus bedrock topography | —— | 500 ×500 m | —— |
| Upward heat flux | Assigns typical values | —— | 500 ×500 m | Flowers et al. (2003); Björnsson. (1988) |
| Ice temperature | Prescribed 0 ℃ everywhere | —— | 500 ×500 m | Schmidt et al. (2020) |

[a] Ji et al. (2014), [b] Collins et al. (2011), [c,d] Watanabe et al. (2011).

Line 78, see comment above, is the daily SMB filed used or monthly as stated in line 60?

Should be monthly, we have corrected it.

Line 82-82, what does "lapse rate approach" mean? Do you correct with a temperature lapse rate? What is the value for the rate?

This is a method that downscales 30 km ERA5 climate fields to 0.025 grid, making them has higher resolution that are capable of capturing the VIC topography, and then as observations in ISI-MIP method to downscale and bias-correct climate from ESM. The lapse rate is calculated by the linear relationship of surface elevation against each climate variable in Yue et al. (2021). We added Section "2.3 SMB modelling" to describe how we downscale the ESM climate fields:

"In this study, the SMB fields used to drive PISM are from Yue et al. (2021), and estimated by SEMIC under the historical, G4, RCP4.5 and RCP8.5 scenarios during 1982–2089. SEMIC in turn is driven by downscaled and bias-corrected ESM data including temperatures, windspeeds, pressures, humidities and radiative forcing terms. We use all CMIP5 and GeoMIP ESM that have complete data fields available, namely BNU-ESM, HadGEM2-ES, MIROC-ESM, and MIROC-ESM-CHEM (Table 1). We statistically downscaled the ESM forcing based on the ERA5 reanalysis dataset (Hersbach et al., 2020). The point of the bias correction is to ensure the mean state of the model parameters matches observations. The separate model trends within each ESM over the observational period remain the same. Thus, the bias correction ensures that models begin close to an observed state, but can then diverge as the separate model climate dictate. The spatial resolution of ERA5 is about 30 km, but still cannot capture the VIC topography. To address this, we first downscaled ERA5 climate to $0.025° \times 0.025°$ grid based on their correlation with VIC surface elevation. We find surface elevation is well correlated with near-surface temperature (R=0.83, p<0.01), downward longwave (R=0.77, p<0.01) and shortwave radiation (R=0.74, p<0.01) and specific humidity (R=0.77, p<0.01), with lapse rates of -5.4 °C $km^{-1}$, -11.9 W $m^{-2}$ $km^{-1}$, 15.85 W $m^{-2}$ $km^{-1}$ and -0.59 k $k^{-1}$ $km^{-1}$, respectively. Precipitation and snowfall are downscaled following De Ruyter-de Wildt et al. (2004). The former is downscaled using Kriging interpolation method, with its empirically exponential relationship with observed surface elevation. The latter is assumed equal to precipitation rate when the daily mean air temperature is below 3°C, otherwise no snowfall occurs. Other SEMIC driven fields (surface wind speed, air density, pressure) are simply bilinearly interpolated due to the relatively minor effects on SMB in SEMIC. Then, we use the downscaled $0.025° \times 0.025°$ forcing fields as the observational reference climate to downscale and bias-correct the ESM fields using the ISIMIP approach (Hempel et al., 2013). The ISIMIP is a trend-preserving approach so that the long-term climate trends in models are preserved, while the mean at each grid cell is matched to observations. There are two fundamentally different ways ISIMIP can do the correction: addition and multiplication, and we follow ISIMIP protocol in deciding which method to use for each meteorological field variable (Hempel et al. 2013). The additive approach is used for most fields preserving, e.g. the absolute changes of the monthly temperature; while the multiplicative method is used for preserving the relative changes for precipitation and radiation. Finally, these $0.025° \times 0.025°$ fields were used to drive the SEMIC model. We also bias-corrected VIC surface albedo and considered SMB-elevation feedback in all simulations (Yue et al., 2021). Over the whole VIC, modelled SMB over the period 1991–2010 (Fig. 1d, Fig. 2) is well correlated (R=0.6, p<0.05) with an interpolated map from 60 measurement sites (Björnsson et al., 2013), although the mean is overestimated by 0.61 m $yr^{-1}$."

Line 83, what does "in reasonable agreement" mean? Some quantification or comparison would be useful here.

Done. We corrected:

"Over the whole VIC, modelled SMB over the period 1991–2010 (Fig. 1d, Fig. 2) is well correlated (R=0.6, p<0.05) with an interpolated map from 60 measurement sites (Björnsson et al., 2013), although
the mean is overestimated by 0.61 m yr$^{-1}$."

Line 86 (figure 1 caption)   A) is not a location map, it only shows the Vatnajökull ice cap not where it
is located in Iceland, suggest to put inset map that shows whole of Iceland and where Vatnajökull is
located in figure 1a), not that one ' is missing in Tungnaárjökull (the second a should be á) , in d) is is
the "annual average"? suggest to clarify
Done. We corrected:

[Figure]

**Figure 1.** Model input data fields. (a) Vatnajökull ice cap (VIC) surface elevation from Spot5 (data
processing methods see Berthier and Toutin, 2008) in summer 2010; (b) bedrock elevation (Björnsson,
1986; Björnsson and Pálsson.2020); (c) ice thickness; (d) applied upward geothermal heat flux (Flowers
et al. 2003), including the Grímsvötn active volcano. (e) annual average surface mass balance 1982-1999
simulated by SEMIC forced by four Earth System Models (Yue et al., 2021). (f) the geographical location
of panel (a, red box) observed by Google Earth.

Line 89 "equilibrium line boundary" is a strange wording, suggest to use the commonly used
"equilibrium line altitude", add something like "applied" or "assumed" before upward geothermal heat
flux
Done. See revisions above.

Figure 1, see comment above, there is space in this figure (lower right corner) to add observed SMB that
would aid the missing comparison with observation (see line 83)
Done. We added the geographical location of VIC in lower right corner, but we added text in quantitative
comparison between modelled and observed SMB:
"Over the whole VIC, modelled SMB over the period 1991–2010 (Fig. 1d, Fig. 2) is well correlated
(R=0.6, p<0.05) with an interpolated map from 60 measurement sites (Björnsson et al., 2013), although
the mean is overestimated by 0.61 m yr$^{-1}$."

Line 91, here it is stated that PISM is forced with monthly SMB fields (see comment line 78), what is the time resolution of the forcing?

It's monthly, we have corrected above errors.

Line 92-93 sentence is strange, something is missing, suggest something like: The final year of the spin-up simulation is then used as the initial condition in the experiments (or scenario simulations).

Done. We followed your suggestion.

Line 96, figure 2 caption, suggest to add "simulation" after spin-up and also state if the forcing is annual, monthly or daily averaged over this period (hat is the time resolution of the forcing?) and also make sure the period is consistent, here it is stated 1982-1999, in Figure 1 the average surface mass balance is shown for the period 1982-2005.

Done. We added "simulation" in figure caption, and we revised figure 1 SMB period to 1982–1999.

Line 97 here it is stated that PISM is forced with 4 different ESM, is then the SEMIC model not used? See comment above, suggest to be consistent in describing the surface forcing method.

SEMIC modelled SMB was used to drive PISM. We revised caption:

"PISM modelled Vatnajökull ice cap (VIC) volume change (a) from the 2000-year climate spin-up simulation driven by repeated monthly SMB fields during 1982–1999 from SEMIC modelling outputs (Yue et al., 2021), driven with downscaled and bias-corrected climate forcings by four Earth System Model."

Line 99, it is strange to show the ensemble mean spatial distribution, as 2 of the models in the 4 piece ensemble have negative and 2 have positive difference, these could therefore cancel out in some location, suggest to either show only one, or all four, so it is possible to assess the performance of each simulation. Line 101, is the magenta line the ensemble mean extent? See comment above, it is more useful to show each model separately.

Done. We showed four ESM separately. The magenta curves represent the extent after spin-up.

[Figure]

**Figure 3.** PISM modelled Vatnajökull ice cap (VIC) volume change (a) from the 2000-year climate spin-up simulation driven by repeated monthly SMB fields during 1982–1999 from SEMIC modelling outputs (Yue et al., 2021), driven with downscaled and bias-corrected climate forcings by four Earth System Model. The equilibrium volume is slightly different than present day by -1.3% for BNU-ESM, -0.5% for HadGEM2-ES, and 0.8% for both MIROC models. Subplots (b–e) are the spatial distribution of VIC thickness differences (ice thickness after spin-up minus present ice thickness) from PISM driven by (from b–e) BNU-ESM, HadGEM2-ES, MIROC-ESM and MIROC-ESM-CHEM. The black curves represent the present ice cap extent. The magenta curves represent the extent after spin-up.

Line 103 suggest to add a reference for SMB-altitude feedback. Add "change" after elevation. See comment above about the period, in caption for Figure 2 the period is stated 1982-1999

Done, we added Edwards et al. (2014), we added "change" after elevation. We would like to keep the reference period as 1982–2005, as we would like to consider the feedback in CMIP5 future scenario.

Edwards, T. L., Fettweis, X., Gagliardini, O., Gillet-Chaulet, F., Goelzer, H., Gregory, J. M., Hoffman, M., Huybrechts, P., Payne, A. J. and Perego, M.: Effect of uncertainty in surface mass balance–elevation feedback on projections of the future sea level contribution of the Greenland ice sheet, Cryosph., 8(1), 195–208, 2014, doi.org/10.5194/tc-8-195-2014

Line 104, suggest to use another word than "correct", It is not clear that the resulting SMB is more correct than the original (how can you assess that?), in equation it is called SMBadj, why not call it then "adjusted" with more explanation?

Done. We changed to:

"We therefore considered the SMB-elevation feedback in annual SMB forcing with the k,"

$$\text{SMB}_t^{\text{adjusted}} = \text{SMB}_t^{\text{SEMIC}} + k \times \left( h_{t-1}^{\text{PISM}} - h_0^{\text{PISM}} \right) \tag{7}$$

Line 105 suggest to use different wording for "ESM-dependent "SMB lapse rate"" suggest to explain better what is meant and define what k is and how it is determined.

K is the gradient of annual mean SMB with observed surface elevation during 1982–2005. We corrected:

"The SMB-elevation feedback (Edwards et al., 2014) alters SMB as VIC topography evolves, and we take this into account in the 2006–2089 simulations. Yue et al. (2021) found VIC surface elevation changes and historical SMB over 1982–2005 were significantly correlated ($R^2$>0.7, p<0.01), and calculated the "SMB lapse rate" *k* (the gradient of annual mean SMB with surface elevation during 1982–2005) in different ESM. We therefore adjust SMB forcing, and ice thickness changes modelled by PISM in the year *t* from 2006 to 2089 as"

Line 109, see comment above, suggest to "adjusted" rather than "corrected"

Done.

Line 110, is this the modelled ice thickness in 2005? In Figure 2 is appears to be in year 1999 why is 2005 selected? See comment above, how is k determined?

The $h_0^{PISM}$ in Line 110 means the modelled ice thickness value is at the end of 2005, the choice of 2005 instead of 1999 is because we just want to consider the SMB-elevation feedback in CMIP5 RCP future scenario. k values are explained in reply of Line105. We corrected the $h_0^{PISM}$ description: "$h_0^{PISM}$ is the modelled ice thickness at the end of 2005". We also added: "We considered SMB- elevation feedback in the CMIP5 future period 2006–2089." in section 2.4, to make the period of feedback correction clearer.

Line 112-113, this text reads awkwardly, suggest to use volume change for the evolution, but here write the difference between steady state and measured, or something like that. Is the average over one year used? From Figure 2 it appears that the seasonal volume change is considerable.

Yes, it is the average over one year (2000), We corrected:

"After the spin-up, VIC volume differences (averaged over 1 year) for the four ESM are between -1.3 % and 0.8 % of measured volume, while the area is around 16 % lower than observed (Fig. 3)."

Line 113 suggest to replace "Ice area loss" with difference between simulated state state and measured, see comment above. Suggest to replace "over" with "at"

Done. We corrected:

"Differences between simulated state and measured are mainly at the outlet glaciers of Dyngjujökull, Brúarjökull and Síðujökull (location see Fig. 1, Fig. 3) where the measured ice thicknesses are less than 100 m."

Line 115, suggest to add "measured" before "ice thickness". Also suggest to use difference between steady state (or spin-up state) and measured, rather than "changes"

Done.

Line 116, this phrasing "are consistent across all the ESM" is strange, suggest to write something like the spin-up steady states forced with the 4 ESM have similar steady-state geometry, or something like that

Done, we changed to "Differences between steady state and measured in VIC geometry are largely determined by the SMB field, and the spin-up steady states forced with four ESM have similar steady-state geometry (Fig 3, b–e)."

Line 117 here is strange wording, suggest to replace "that drive SMB" with something mentioning SEMIC model. Here is for first time the "bias-correction with ERA5 reanalysis mentioned, it should be clearer before that the all the ESM are "bias-corrected" with the same data.

Done. We corrected "This because all climate variables (e.g., surface air temperature, downward longwave and shortwave radiation) that drive SEMIC model are bias-corrected with ERA5 reanalysis using ISIMIP approach (Section 2.3)."

In line 82 it is stated that ESM were bias corrected using ISI-MIP.   What does that actually mean? Are the annual or monthly averaged added or subtracted from the ESM values?

ISI-MIP is a trend-preserving approach so that the long-term climate trends in models are preserved, while the mean at each grid cell is matched to observations. There are two fundamentally different ways ISIMIP can do the correction: addition and multiplication. The additive approach is used for most fields preserving e.g. the absolute changes of the monthly temperature; while the multiplicative method is used for preserving the relative changes for precipitation and radiation. We added a brief description of ISI-MIP in Section 2.3

"The ISIMIP is a trend-preserving approach so that the long-term climate trends in models are preserved, while the mean at each grid cell is matched to observations. There are two fundamentally different ways ISIMIP can do the correction: addition and multiplication, and we follow ISIMIP protocol in deciding which method to use for each meteorological field variable (Hempel et al. 2013). The additive approach is used for most fields preserving, e.g. the absolute changes of the monthly temperature; while the multiplicative method is used for preserving the relative changes for precipitation and radiation."

Line 118-124 this whole explanation is very confusing, suggest editing the whole paragraph. The discrepancies are not caused by surging glaciers, the fact that most of the outlet glacier of Vatnajökull on the north and western side are surging and the model does not include any surging could be the reason for the model failing in simulating the observed ice thickness, that should be made clearer in this paragraph. Suggest to take out "not parameterized" and use something like, not modelled or not included.

We revised:

"The largest discrepancies between the spin-up and present area for VIC, are likely due to surge type glaciers, which is not a process simulated by PISM. Many glaciers on the northern and western sides of VIC are of surge type (Björnsson et al., 2003), and this is where differences in observed ice thickness and in PISM are largest. Surges rapidly move long-accumulated ice from the upper glacier towards the terminus, so that at any particular time the upper and lower glacier are not in the average state that PISM simulates. Thus, the spin-up is unlikely to achieve a present-day area coverage, although total volume is close to observed."

**3 Ice cap volume and area from 1982 to 2089**

Line 127, In Table 1 only 2089 relative to 1982 is shown, not the difference duing 1991-2014, was that intended?

No, the changes 1991-2014 are shown in Table 1 (in bold here).

**Table 1.** Vatnajökull ice cap volume and area change (%) during 1991–2014 (volume during 2006–2014 is the mean of RCP4.5 and RCP8.5 scenarios), and 1982–2089 under G4, RCP4.5 and RCP8.5 scenarios modelled by PISM forced by BNU-ESM, HadGEM2-ES, MIROC-ESM, MIROC-ESM-CHEM, the ensemble mean and 95% confidence intervals, N=4. Numbers in brackets represent changes without considering SMB-elevation feedback.

| | | BNU-ESM | HadGEM2-ES | MIROC-ESM | MIROC-ESM-CHEM | Ensemble |
|---|---|---|---|---|---|---|
| Volume | G4 | 14 (13) | 10 (10) | 11 (11) | 13 (12) | 12±2 |
| | RCP4.5 | 21 (20) | 18 (17) | 11 (11) | 13 (13) | 16±4 |
| | RCP8.5 | 25 (23) | 23 (22) | 20 (20) | 22 (21) | 22±2 |
| | **1991-2014** | **2** | **2** | **0** | **0** | **1±1** |
| Area | G4 | 10 (9) | 6 (6) | 8 (7) | 9 (8) | 8±2 |
| | RCP4.5 | 14 (12) | 11 (11) | 8 (7) | 9 (9) | 10±3 |
| | RCP8.5 | 15 (14) | 14 (14) | 12 (12) | 13 (12) | 14±1 |

Line 127-128 neither the overestimation of SMB nor the disappearance of fast melting region are shown, more explanation is needed here.

OK, we rephrase this:

"Pálsson et al. (2015) record a 3% reduction in volume between 1991–2014 which is more than the 1±1 %

(we define uncertainties in this study as the ensemble mean and 95% confidence interval, N=4) we simulate (Table 1). This is due both to the VIC SMB used to force PISM being overestimated by 0.61 m yr$^{-1}$ compared with the interpolated map from 60 site measurements during 1991–2010 (Björnsson et al.,

2013), and also the rapid loss of area during the model spin up which removed the thin and fast melting regions at Dyngjujökull and Brúarjökull (Fig. 1, Fig. 3, b–e)."

Line 132, suggest to edit this sentence, it is very vague and more quantification and comparison would be useful, "likely reason" and "somewhat difference ice cap geometry" could be made clearer or better quantified.

We quantitively showed the difference between observed and steady state VIC geometry at eastern outlet glaciers:

"However, there are some large differences mainly over the eastern outlet glaciers where PISM

overestimates the velocity by more than 100 m yr$^{-1}$. This is related to VIC surface elevations being 50–

150 m lower than measured at eastern outlet glaciers (Fig. 3)."

Line 135-137 suggest to edit the whole figure caption and reconsider the ensemble and scenario averaged, suggest to show only one, or maybe two (there is space in the figure for at least, if not 3 more subfigures).

The text is redundant in two places "RCP4.5 and RCP8.5" are two times in same sentence and "average"

and "mean", suggest to delete one of the two occurrences.

We showed separate model results under RCP4.5, instead of model and scenario mean. We changed:

[Figure]

Figure 4. Top: Mean surface velocity over VIC from Sentinel-1, 100 m spatial resolution product (Wuite et al., 2021). Middle row: mean 2015–2020 surface velocities simulated by PISM under the RCP4.5

scenario from the 4 Earth System Model as labeled. Bottom row: the PISM-Sentinel differences.

Line 137 suggest to replace "spaced" with "spatial resolution" and replace (upper left) with (upper right)

Done.

Line 139 see comment above Table 1 does not show historical changes as stated in lines 126 ad 140.

It does, see reply of Line 127.

Line 141 are those 12% and 22% values relative to initial (which?) or maximum volume? It is not clear from text

It's during 1982-2089, so, the volume change is relative to 1982, we corrected:

"During the period 1982–2089, annual volume loss and SMB are well correlated in all ESM (Fig. S1;

R=0.98, p<0.01). The across-ESM ensemble mean of VIC volume loss is decreased by as little as 12 %

under G4 to as much as 22 % under RCP8.5."

Line 142, add "loss" after "volume"

Done.

Line 144 missing 'over second a in Tungnaárjökull

Corrected.

Line 145 This statement is not correct as shown in the 4th row of figure 5 for both the MIROC simulations, the difference is 0 (negative values are not shown, if there are any?) and the volume and area loss of G4

and RCP4.5 are very similar as shown in Figure 4

Yes, it has some negative values, but could be ignored. We changed the colorbar of Figure 4, the scale is from -10 to 110 m. We corrected:

"Surface thinning under G4 is smaller than that under RCP4.5 in BNU-ESM and HadGEM2-ES, while two MIROC models display negligible differences (<5 m) in surface elevation."

[Figure]

Figure 6. The ice thickness differences from PISM outputs between the year 1982 and 2020 over Vatnajökull ice cap under G4 (1st row), RCP4.5 (2nd row) and RCP8.5 (3rd row) scenarios, and their differences (G4-RCP4.5, 4th row; G4-RCP8.5, 5th row) by Earth System Model (ESM, from left to right), BNU-ESM, HadGEM2-ES, MIROC-ESM, MIROC-ESM-CHEM and ensemble mean. The initial state in 1982 is different for each ESM.

Line 145-146 this statement of G4 increasing ensemble ice thickness is strange, see comment above about ensemble mean not being useful, and that G4 increasing thickness is not true, the response of the model when G4 is that the thinning of the ice cap is reduced.

We disagree about the ensemble mean being useful, but we rewrote:

"By 2089, all four ESM simulations under all scenarios produce surface thinning over the whole VIC especially over Tungnaárjökull, Brúarjökull (location see Fig. 1) and eastern small outlet glaciers (Fig. 5). Surface thinning under G4 is smaller than that under RCP4.5 in BNU-ESM and HadGEM2-ES, while two MIROC models display negligible differences (<5 m) in surface elevation."

Line 149 see comment above, the ensemble mean is really not useful here, as it is taking the attention away from the interesting differences in the model responses.

We disagree that the ensemble mean is useful, however that is not relevant to this line which describes a figure with all the separate models plotted as well as the ensemble mean. We also added the description in Section 3 to emphasize the individual models results.

Line 150 suggest to replace "Estimates considering ice dynamic from PISM" with "volume and area loss simulated by including ice dynamics"

Done.

Figure 5 in top line MIROC-ESM is misspelled as MIROE. The two bottom line figures should be shown with the same color scale for aiding comparison it is misleading to show differences with same color scale but different values, suggest to have both scales go to 100 m so that for example yellow color doesn't show 50 m in one and 70 m in the other row.   It is not clear (figure caption states ice thickness differences between 2089 and 1982 is it the same initial state or ESM specific 1982 state? How different are the initial states at 1982?

We corrected the "MIROC" and used the same color scale. The initial state is ESM specific state, but each ESM state is very similar (Figure 3). We revised the color bar in figure as suggested:

[Figure]

Figure 6. The ice thickness differences from PISM outputs between the year 1982 and 2020 over

Vatnajökull ice cap under G4 (1st row), RCP4.5 (2nd row) and RCP8.5 (3rd row) scenarios, and their differences (G4-RCP4.5, 4th row; G4-RCP8.5, 5th row) by Earth System Model (ESM, from left to right),

BNU-ESM, HadGEM2-ES, MIROC-ESM, MIROC-ESM-CHEM and ensemble mean. The initial state in 1982 is different for each ESM.

Table 1 In this table no historical differences are shown as stated in Lines 127 and 139 (see comments above).   See comments above that the ensemble mean with 4 ensemble members is not useful here. This table shows that very little difference is between the runs that couple ice dynamics with the SMB and the runs that have only SMB, therefore the statement in abstract line 18 seems an overestimate, how is ¼ and 1/3 difference found?

The historical period loss is in the table as shown in reply to the earlier questions where we reproduced the table. We disagree about the ensemble mean being useful, as this is common practice, e.g. by IPCC, and in any case the separate ESM are also listed in the table, so removing the ensemble mean would only make the table less informative than at present. The numbers in brackets are without considering SMB-elevation feedback, and **not** the volume changes only caused by SMB. We show how the non-SMB component is derived in Section 4: Ice cap SMB, MB and non-SMB from 2020-2089, which is immediately after this table, and we show where the 1/4-1/3 factors arise. However, we revised the abstract to make the description clearer:

"All ESM show that the non-SMB component (i.e., ice dynamics and basal melting) remains nearly constant at around -0.25 m yr$^{-1}$ and is remarkably insensitive to climate forcing over time for all scenarios. This non-SMB component is important for ice cap loss rates compared with mass balances of -0.47, -0.61 and -0.88 m yr$^{-1}$ over the 1982–2089 period under G4, RCP4.5 and RCP8.5, respectively."

**4 Ice cap SMB, MB and non-SMB from 1982 to 2089**

Line 163 "with maximum of more than 400 m" this seems large, given the mean thickness of the ice cap. Over how long period? What are the velocities that move this accumulated mass?   Is this realistic or not?

It's over the 1982–2089, as is said fig.6 caption. Fig. 1d shows that in the area with maximum height gain, the SMB is 6-8 m/yr. Over a century this plausibly can explain why the SMB can raise elevations by 400 m. Velocities are given in revised Fig. 4 both from Sentinel and PISM as shown above. We rewrite the section more explicitly:

"In Fig. 7, we separate the SMB and non-SMB (ice dynamics and basal melting) components of overall mass balance. Over the 1982–2089 period, simulated SMB decreases the average ice thickness of the whole VIC by 40–80 m especially over the outlet glaciers of Skeiðarárjökull and Breiðamerkurjökull (location see Fig. 1) while increasing the ice thickness over the interior of VIC, by a maximum of 400 m over the southern region of VIC where mass balances are highest (Fig. 1e). There is a larger area of surface thinning region under RCP8.5 than under RCP4.5 and G4 scenarios due to the higher air temperatures (Yue et al., 2021)."

Line 165-166 suggest to edit, "the smallest area of surface thinning" is strange wording. Also given the known higher temperature in RCP8.5 it is not surfacing that surface thinning is stronger for that scenario, by how much? Is even over the ice cap? Is it realistic differences? Why is there so little difference between RCP4.5 and RCP8.5 in the MIROC simulations?

We corrected:

"There is a larger area of surface thinning region under RCP8.5 than under RCP4.5 and G4 scenarios due to the higher air temperatures (Yue et al., 2021)."

 this sentence "Non-SMB components display the opposite pattern to SMB" should be deleted, it indicates little understanding of dynamics of ice cap.

Done.

 suggest to delete or edit this sentence to include ice dynamic understanding as it is written is seems like authors are analysing model results that are little understood.

We have rewritten these sentences to be clearer:

"Positive non-SMB contributions to mass balance are visible in all ESM and scenarios around the margins, because as the negative SMB reduces surface elevation in the margins, the surface gradient between the interior and the margins in increased, driving an increased ice flux into the margins. Conversely, this increased ice flux removes mass from the interior, making the non-SMB component there negative."

 See comments above, the interesting results are that there is difference between the responses of the different ESM forcings, giving numbers for the ensemble (and showing in Figure 6) is hiding these interesting results.

We added descriptions for individual ESM.

"Basal melting is driven by non-climate factors and so remains essentially unchanged under the scenarios. The pattern of non-SMB contributions for individual ESM are all quite similar, the largest differences being mainly over the ablation zone, with the across-model standard deviations more than 10 m (Fig. S2).

Fig. 7b demonstrates that surface height differences (G4-RCP4.5) by 2089 are mainly caused by SMB rather than non-SMB effects. Ensemble mean SMB under G4 increases VIC mean surface height by around 20 m than RCP4.5 scenario, largely due to BNU-ESM and HadGEM2-ES, while the difference is less than 10 m for both MIROC models (Fig. S3–S6). For G4-RCP8.5, SMB driven height differences under HadGEM2-ES are moderately greater than for BNU-ESM, and much greater than in two MIROC models, especially at Tungnaárjökull, Dyngjujökull and Brúarjökull (location see Fig.1). G4 dynamically thickens the ablation zone relative to the RCP scenarios, while thinning the accumulation area. The dynamic impact on surface height differences between G4 and RCP4.5 is much less between G4 and RCP8.5 (Fig. 7b). Surface height differences (G4-RCP4.5) by non-SMB in both MIROC models are 0–5 m, notably less than that in BNU-ESM and HadGEM2-ES."

.

 analysing the ensemble mean really hides the results shown in Figure 4, suggest to focus on that, rather than the ensemble mean with such small number of members and varying responses.

Agreed:

"The non-SMB contributions, however, remain nearly constant (around -0.25 m yr$^{-1}$) over time across all scenarios and ESM (Fig. 7, Fig. S3–S6). These are fairly large fractions of total ice cap loss rates, but diminish in relative size as MB becomes more negative from -0.47 m yr$^{-1}$ during 1982–2089 under G4, to -0.61 m yr$^{-1}$ under RCP4.5, and -0.88 m yr$^{-1}$ under RCP8.5. Simulations under individual ESM are shown in Fig S3–S6, the responses of MB to G4 and RCP scenarios are very similar to changes in ice cap volume and SMB. MB has the smallest differences (G4-RCP4.5) for the two MIROC models, but relatively large differences for BNU-ESM and HadGEM2-ES."

Figure 6 See comment above about the ensemble mean, the different responses between the 4 ESM is really interesting and that is lost in this figure that only shows the means and therefore misleading. Here the reference is year 2020 but both in Figure 5 and Table 1 the reference year is 1982, why not have the same reference in all figures and table?

We added the descriptions about the individual model results, and we changed period to 1982–2089 that consistent with above figure.

In figure 6b) large difference is between the dynamic (here called (dynamic), in (a) it is called (non-SMB), suggest to be consistent). How can the dynamic part be so different with same ice dynamic model? Figure 7 shows that the non-SMB part is very similar for all simulations, this figure is really strange showing such a large difference. The difference between G4 and RCP4.5 is very small, but Figure 4 shows that each of the ESM has very different response.

We changed the label to "non-SMB" instead of "dynamic". See reply to next comment for more.

Line 192-196 see comment above, suggest to discuss separately each ESM response, as shown in Figure 7, than the mean. The large 95% confidence interval with N=4 clearly shows how variable the responses are.

We changed:

"During the SAI G4 implementation period 2020–2069, G4 increases ensemble mean MB by between $0.21\pm0.17$ m yr$^{-1}$ (95% confidence intervals; N=4) compared with RCP4.5 and by $0.33\pm0.22$ m yr$^{-1}$ compared with RCP8.5, which are very similar to the SMB differences of $0.20\pm0.16$ m yr$^{-1}$ (G4-RCP4.5) and $0.31\pm0.21$ m yr$^{-1}$ (G4-RCP8.5). These numbers demonstrate that the extra ice mass preserved under SAI is through the increases of SMB, rather than non-SMB components, especially in BNU-ESM and HadGEM2. The two MIROC models project almost no differences in both MB and SMB between G4 and RCP4.5, and so is again consistent with the domination of SMB in changing MB, and the unchanging magnitude of the non-SMB component. The SMB and MB under G4 have much larger across-ESM differences than between the two RCP scenarios, due the differences of G4 atmospheric forcings between each ESM (Yue et al., 2021)."

**5 Discussion**

Line 202-203 this sentence could be more clear, the non-SMB appears to have similar value throughout, which I think is clearer information than the the fraction becomes less important.

We changed:

"During the historical period, our simulations show the overall mass loss on VIC is about equally divided between SMB and non-SMB components, but as SMB becomes more negative, the proportion of MB due to non-SMB becomes less, as non-SMB component remains constant over the whole simulation period (Fig. 7c)."

Line 207 this is strange, what about the impact on precipitation or temperature? I would think that it directly the forcing that impacts the response, rather than the degree of imbalance, could you confirm?

Yes, the actual ice mass loss in HMA depends on the forcing, the point of this sentence is about the relative efficacy. We added that forcing is of course important.

"The differences in efficacy from VIC to HMA are related not only to the climate forcing differences between scenarios, but also to the degree of imbalance of the ice masses in present and recent climate, with most of HMA losing ice mass throughout the last century, so losses by 2069 under RCP4.5 are 73%, and under G4 59%, of present-day glacier mass. Iceland has been much closer to balance until recently."

Line 209 "Iceland has been closer to balance until recently" is not very clear, what is recent here? The glaciers in Iceland were close to balance in period 1960-1995, after 1995 the mass balance became negative, and the rate of mass loss reduced after 2010.

Yes, this is much close to balance than HMA has been throughout the 20$^{th}$ century. We added "much":

"Iceland has been much closer to balance until mid-1990s."

Line 211 it is strange to discuss the relative effectiveness of SAI on reducing surface runoff, what is the effect on precipitation, temperature, atmospheric circulation?

Actually, the surface runoff should be called "surface-melted runoff" that is largely determined by melting water, so, it is a key variable to reflect the changes of temperature. We replace it with "surface-melted runoff". We added one paragraph to describe the geoengineering effect on precipitation, temperature:

"In G4, changes in Atlantic Ocean circulation may increase VIC temperatures. Projections by all ESM with data show AMOC index at 30°N is 0–4 Sv stronger in G4 than RCP4.5 (Fig. 9a), which acts to increase heat flux from ocean to atmosphere near Iceland (Fig. 9d). However, the atmospheric cooling associated with G4 SAI dominates the VIC climate, resulting in a 0.4°C reduction of air temperature and a 6% lower surface melt-runoff under G4. There are across model differences, with the two MIROC projecting little changes between G4 and RCP4.5 in temperatures and precipitation, and hence the response of ice cap volume. Precipitation is the main component of mass accumulation, all ESM project insignificant precipitation differences between G4 and RCP4.5. This is different from the global (Trisos et al., 2018) and Greenland (Moore et al., 2019) cases where G4 reduces precipitation in most regions, due to the fundamental difference between long wave greenhouse gas and shortwave SAI radiative forcing. Greenhouse gases are distributed throughout the atmosphere, while shortwave radiation impacts surface temperatures, hence temperature lapse rates are altered under SAI and the atmosphere is drier than it would be for the same temperature under simple greenhouse gas climates. The changes precipitation under G4 that are seen in VIC may be driven by the relatively enhanced AMOC and lower Arctic sea ice (Xie et al., 2022) which in turn brings more water vapor to VIC."

Line 212 It is not clear what the "compensating impact of AMOC changes" are here, the correlation between AMOC and SMB is shown, but what are the physical relationship? (what effect of precipitation and temperature are caused by AMOC changes?) this needs more discussion

See the previous answer, which address how the AMOC brings warmth to Northern Atlantic regions.

Line 219 what is "SMB behavior" clarification is needed

The SMB behavior is the correlation between SMB and AMOC. We changed to:

"Fig. 9b-c shows that VIC MB is highly significantly correlated with AMOC (R=0.91, p<0.01), while for Greenland there is no significant relationship (R=0.42, p=0.35), consistent with the SMB response to

AMOC over VIC and Greenland (Yue et al., 2021)."

Line 222 the sentence "may induce larger dynamic effects earlier" is not clear, needs editing. The dynamic effect appears to be very similar throughout the simulations as shown in Figure 7

We mean that dynamic effects would be expected earlier than in Greenland, but yes, they are not seen in the 50 year SAI period considered here. We clarified this sentence:

"Because VIC is much thinner than the Greenland ice sheet, and has higher accumulation and ablation rates, the mass turnover time in VIC is at least 10 times faster than in Greenland meaning that surface climate may induce larger dynamic effects on centennial timescales."

Figure 8 Why are now 8 different ESM shown? Why are not all included in the analysis earlier in the paper?

Because for the G4 experiment only the 4 ESM that we analyzed in this paper have sufficient data available for SEMIC. The other 4 ESM we used is only to show the poor correlation between AMOC and GrIS mass balance. We stress the only 4 ESM available for geoengineering G4 experiments:

"The SMB fields are modelled by a mass and energy balance model Section 2.1 and 2.3) driven by downscaled and bias-corrected climate forcings by all Earth System Model (ESM; Table 1) that have sufficient data fields available from both RCP and G4 scenarios."

Line 228 "annual mean maximum" is strange here, how is it both mean and maximum?

Should delete 'mean', We changed to:

"The AMOC index is defined as the annual maximum of the overturning stream function over the Atlantic Ocean at 30°N"

Line 236 "effects might be expected to be rather too small to be seen" is strange here, suggest to edit section and clarify

We deleted the word "rather"

Line 239 something is missing "changing elevation-SMB" add "feedback"?

Yes, done.

Line 242 not clear why "extreme maritime environment" (what is extreme about it?) makes a glacier most likely to exhibit a dynamical response, suggest to edit and clarify and also why such an effect I not seen in the experiment in this study.

We mean that it is a modestly small ice cap adjacent to the North Atlantic Ocean and so much closer to the open sea that even those on Arctic archipelagos where seasonal sea ice covers the ocean for parts of the year. As noted earlier for the Line 39 comment on maritime glacier sensitivity to climate change (Oerlemans, 1992; Rupper and Roe, 2008). We make this more explicit:

"The environment of VIC is close to open seas year-round, in contrast with the seasonally ice-covered waters near Vestfonna. Maritime glaciers tend to be more sensitive to climate that more continental ones (Oerlemans, 1992; Rupper and Roe, 2008), and so might be expected to exhibit a dynamical response to the SAI or RCP scenarios, but we see no such effect."

Line 246 The sentence "Furthermore, retreat of the margins from the ocean" is not right here, there are no outlet glaciers of Vatnajökull residing in the ocean, the Jökulsárlón is inland lagoon, connected to the

| 1103 | ocean by a river, but it is not ocean. |
| --- | --- |
| 1104 | We corrected to: |
| 1105 | "Furthermore, calving is confined to just the inland Jökulsárlón lagoon (location see Fig. 1)." |
| 1106 | |
| 1107 | Line 251-251 sentence is strange and no connection between first and second part of it, suggest to edit. |
| 1108 | We revised to: |
| 1109 | "Some previous simulations of VIC had difficulty establishing present-day steady-state geometries in |
| 1110 | spin-up simulations (Aðalgeirsdóttir et al., 2005; Marshall et al., 2005; Flowers et al., 2005). Our |
| 1111 | modelled steady state VIC geometry is similar as observations, with only ±1% differences in ice volume. |
| 1112 | Our projections by 2089 show smaller losses (16±4% for RCP4.5, and 22±2% for RCP8.5) than the e.g. |
| 1113 | 30% loss under RCP4.5 in Flowers et al. (2005). Perhaps unsurprisingly our results are consistent with |
| 1114 | Schmidt et al. (2020), with a 17% volume loss under for RCP4.5, given that we use the same ice dynamic |
| 1115 | model although with different SMB forcing. This leads to local differences in steady state ice thickness." |
| 1116 | " |
| 1117 | Line 255 suggest to edite "in various basin ice thicknesses by 2089" does not make sense here |
| 1118 | Changed to "This leads to local differences in steady state ice thickness." |
| 1119 | Line 258 what does "the relatively paramterized SEMIC model" mean, suggest to clarify |
| 1120 | Changed to: "especially in the SEMIC model, which uses parameterizations established in Greenland" |
| 1121 | Line 259 suggest to edit "is still not perfectly captured" better to quantify, would you expect perfect |
| 1122 | capturing? When? |
| 1123 | Changed to: The steep geometry of some outlet glaciers is not fully resolved by the 0.025°×0.025° (about |
| 1124 | 1.2 km ×1.2 km) grid although the bias-correction using satellite observations of albedo corrects offsets |
| 1125 | from model to observations. |
| 1126 | |
| 1127 | Line 260-261 strange sentence suggest to edit and clarify, not clear hoe albedo compensates for resolution? |
| 1128 | Bias correction serves to correct errors in mean state, so the relative lack of resolution of steep slopes |
| 1129 | can be compensated for by the bias correction ensuring the mean matches the observations. We corrected: |
| 1130 | "The steep geometry of some outlet glaciers is not fully resolved by the 0.025°×0.025° (about 1.2 km |
| 1131 | ×1.2 km) grid although the bias-correction using satellite observations of albedo corrects offsets from |
| 1132 | model to observations." |
| 1133 | |
| 1134 | Line 265 what is "de-weighting" suggest to edit |
| 1135 | We revised: |
| 1136 | "Moore et al. (2019) evaluated de-weighting each MIROC model in ensemble Greenland simulations; |
| 1137 | reducing each MIROC model contribution to the ensemble mean by 25% made little difference to the |
| 1138 | equal-weight ensemble means, and in general, the two ESM are considered independent in climate |
| 1139 | simulations." |
| 1140 | |
| 1141 | Line 265-268 strange sentences and suggest to edit, it is speculative "could perhaps provide improved |
| 1142 | polar impact studies |
| 1143 | The sentence is essentially true since no one yet has published results with polar G6 impacts. Changed |
| 1144 | to: The new generation of ESM that participated in CMIP6, and with new corresponding GeoMIP G6 |
| 1145 | experiment are slowly becoming available and might improve polar impact studies. |
| 1146 | |

Line 270 what does "not particularly effective" mean?

It is relative to geoengineering impacts in Greenland ice sheet. We corrected to:

"Although geoengineering by SAI is not as effective for VIC as Greenland, it does still slow the rate of
ice loss."

Line 271 "unique geographical location" is strange, isn't every location unique?  "we may infer" is
strange here, suggest to delete

Yes, deleted unique.

Line 272 sentence is strange "will not lead to greater mass loss of any glacier of ice cap" suggest to edit
or delete

We corrected:

"The North Atlantic and maritime setting VIC makes it potentially more susceptible to the warming
impacts from AMOC under G4 than other Arctic ice caps. However, this study demonstrates that SAI as
specified by G4 will not lead to greater mass loss at VIC, and by extension, of any glacier or ice cap in
the northern hemisphere, than are expected under any plausible greenhouse gas scenario."

Line 274-275 suggest to delete.  What is "palatable governance issues"?   Moore et al., 2020 is not in
reference list

Governance issues for SAI are very controversial and well explored in the literature. The topic is
relatively important here since one reason to explore the impacts of SAI is that is a reasonable chance of
it being done. The governance differences between localized innervations and SAI are discussed in
Moore, J. C., Wolovick, M., Gladstone, R., Chen, Y., Kirchner, S. and Moore, J. C.: Targeted
Geoengineering: Local Interventions with Global Implications, Global Policy, 12(S1), 108-118, 2020,
doi:10.1111/1758-5899.12867

**6 Conclusion**

Line 278 "reduces VIC mass loss by 4 percentage points" is strange, why not 4% ? suggest to edit

Because a percentage of a percentage is ambiguous. The standard way of describing a change e.g. from
8% to 4% is to say a reduction of 4% points rather than saying a reduction of 50% (from 8% to 4%).

Line 279 "SAI could help preserve VIC from melting" is not true, the melting of the ice cap happens also
in G4 simulations (suggest to replace "melt" with "mass loss" melting happens every summer)

Done.

Line 281 "compensating changes in temperature and accumulation due to AMOC" is not discussed before
and should be better explained earlier in paper

This is now discussed more fully earlier e.g.: In G4, changes in Atlantic Ocean circulation may increase
VIC temperatures. Projections by all ESM with data show AMOC index at 30°N is 0–4 Sv stronger in
G4 than RCP4.5 (Fig. 9a), which acts to increase heat flux from ocean to atmosphere near Iceland (Fig.
9d). However, the atmospheric cooling associated with G4 SAI dominates the VIC climate, resulting in
a 0.4°C reduction of air temperature and a 6% lower surface melt-runoff under G4. There are across model differences, with the two MIROC projecting little changes between G4 and RCP4.5 in temperatures and precipitation, and hence the response of ice cap volume. Precipitation is the main component of mass accumulation, all ESM project insignicant precipitation differences between G4 and RCP4.5. This is different from the global (Trisos et al., 2018) and Greenland (Moore et al., 2019) cases where G4 reduces precipitation in most regions, due to the fundamental difference between long wave greenhouse gas and shortwave SAI radiative forcing. Greenhouse gases are distributed throughout the atmosphere, while shortwave radiation impacts surface temperatures, hence temperature lapse rates are altered under SAI and the atmosphere is drier than it would be for the same temperature under simple greenhouse gas climates. The changes precipitation under G4 that are seen in VIC may be driven by the relatively enhanced AMOC and lower Arctic sea ice (Xie et al., 2022) which in turn brings more water vapor to VIC.

Line 283 "VIC is relatively insensitive to climate scenario" does not make sense here, suggest to edit or delete

Rephrased "mean that the mass balance of VIC is much less dependent on climate scenario than glaciers in many other regions."

Line 283 "relatively unaffected by changing air and ocean temperature" is not clear, ocean temperature does not affect dynamics as VIC is not in connection to ocean and the results of the study show that the dynamics is affected through changes in geometry of the ice cap. Suggest to edit or delete.

Iceland is surrounded by the sea. AMOC changes ocean temperatures and has an impact on the climate of Iceland, making it much milder than places at the same latitude. Hence ocean temperatures in this sentence:

"We find that ice dynamics are almost constant over both time and scenario because they are relatively unaffected by changing air and ocean temperatures."

Line 384 the paper by Schmidt et al is now published and this reference should be replaced by the Cryosphere paper

Done.

Line 388 two places there should be ð instead of o: Aðalgeirsdóttir and Guðmundsson

Done.

---

## Author Comment (AC2)

**Reviewer 2**

The main idea seems to be that: (a) Geoengineering reduces the effect of global warming, somewhat, on the Iceland ice cap, and (b) this is different from how Greenland behaves for a variety of reasons: thinner, smaller, fewer marine terminating glaciers, high precipitation, high geothermal heat flow, etc. HOWEVER... the main ideas are repeatedly lost in a mountain of detail. Needs major rewrite to put the main ideas (whatever they might be) front and center, and use the details to support it.

Figures need major work. They are unnecessarily hard to interpret. Difficult / inappropriate color scales, lack of titles on axes, overall plot and color scale where they could have been used, Hard-to-distinguish lines with fine gradations in colors, large matching problems between lines and legends, lack of glacier labels (except in one figure), etc. This is making the reader work WAY too hard to understand the data in these figures.

The procedure seems pretty straightforward: run an ESM, use it to force PISM, see what happens. And before that, spinup for a certain period of time. Again, this section needs to be better organized around main ideas, rather than getting lost in the details.

Thanks, we have improved our manuscript significantly according to your valued suggestions. We have added a new section (2.3 SMB modelling) to explain the statistically downscaling method more detailed, and describe how we modelled SMB, and detailed information about PISM parameterizations we used, and a table (Table 1, see reply 76-77) about PISM boundary forcing. These all make our method clearer.

2.3: SMB modelling:

"In this study, the SMB fields used to drive PISM are from Yue et al. (2021), and estimated by SEMIC under the historical, G4, RCP4.5 and RCP8.5 scenarios during 1982–2089. SEMIC in turn is driven by downscaled and bias-corrected ESM data including temperatures, windspeeds, pressures, humidities and radiative forcing terms. We use all CMIP5 and GeoMIP ESM that have complete data fields available, namely BNU-ESM, HadGEM2-ES, MIROC-ESM, and MIROC-ESM-CHEM (Table 1). We statistically downscaled the ESM forcing based on the ERA5 reanalysis dataset (Hersbach et al., 2020). The point of the bias correction is to ensure the mean state of the model parameters matches observations. The separate model trends within each ESM over the observational period remain the same. Thus, the bias correction ensures that models begin close to an observed state, but can then diverge as the separate model climate dictate. The spatial resolution of ERA5 is about 30 km, but still cannot capture the VIC topography. To address this, we first downscaled ERA5 climate to $0.025° \times 0.025°$ grid based on their correlation with VIC surface elevation. We find surface elevation is well correlated with near-surface temperature (R=0.83, p<0.01), downward longwave (R=0.77, p<0.01) and shortwave radiation (R=0.74, p<0.01) and specific humidity (R=0.77, p<0.01), with lapse rates of -5.4 °C km$^{-1}$, -11.9 W m$^{-2}$ km$^{-1}$, 15.85 W m$^{-2}$ km$^{-1}$ and -0.59 k k$^{-1}$ km$^{-1}$, respectively. Precipitation and snowfall are downscaled following De Ruyter-de Wildt et al. (2004). The former is downscaled using Kriging interpolation method, with its empirically exponential relationship with observed surface elevation. The latter is assumed equal to precipitation rate when the daily mean air temperature is below 3°C, otherwise no snowfall occurs. Other SEMIC driven fields (surface wind speed, air density, pressure) are simply bilinearly interpolated due to the relatively minor effects on SMB in SEMIC. Then, we use the downscaled $0.025° \times 0.025°$ forcing fields as the

observational reference climate to downscale and bias-correct the ESM fields using the ISIMIP approach (Hempel et al., 2013). The ISIMIP is a trend-preserving approach so that the long-term climate trends in models are preserved, while the mean at each grid cell is matched to observations. There are two fundamentally different ways ISIMIP can do the correction: addition and multiplication, and we follow ISIMIP protocol in deciding which method to use for each meteorological field variable (Hempel et al. 2013). The additive approach is used for most fields preserving, e.g. the absolute changes of the monthly temperature; while the multiplicative method is used for preserving the relative changes for precipitation and radiation. Finally, these 0.025°×0.025° fields were used to drive the SEMIC model. We also bias-corrected VIC surface albedo and considered SMB-elevation feedback in all simulations (Yue et al., 2021). Over the whole VIC, modelled SMB over the period 1991–2010 (Fig. 1d, Fig. 2) is well correlated (R=0.6, p<0.05) with an interpolated map from 60 measurement sites (Björnsson et al., 2013), although the mean is overestimated by 0.61 m yr$^{-1}$."

We rewrote the Results section, and added the description of the 4 ESM responses to climate scenarios, which made our story more complete. We also did lots of revisions to the Introduction, Discussion and conclusion parts, especially on the relationship between AMOC and Iceland and geoengineering. Please see our new revised manuscript.

We redraw almost all figures according your nice suggestions, we changed colorbar scale to discrete segments, rather than the high resolution essentially continuous ones that we prefer as they are better able to see differences in response without threshold values, but we accept your suggestion. As for glacier labels, we think only the labeles in Figure 1 are enough, since these are the only ones which occur in the text, and we refer to Fig. 1 everytime they are used in the text. e.g. "Differences between simulated state and measured are mainly at the outlet glaciers of Dyngjujökull, Brúarjökull and Síðujökull (location see Fig. 1; Fig. 3)"

Here are our revised figures:

[Figure]

**Figure 1.** Model input data fields. (a) Vatnajökull ice cap (VIC) surface elevation from Spot5 (data

processing methods see Berthier and Toutin, 2008) in summer 2010; (b) bedrock elevation (Björnsson, 1986; Björnsson and Pálsson.2020); (c) ice thickness; (d) applied upward geothermal heat flux (Flowers et al. 2003), including the Grímsvötn active volcano. (e) annual average surface mass balance 1982-1999 simulated by SEMIC forced by four Earth System Models (Yue et al., 2021). (f) the geographical location of panel (a, red box) observed by Google Earth.

We also showed separate model results, instead of ensemble mean:

[Figure]

**Figure 3.** PISM simulated Vatnajökull ice cap (VIC) volume change (a) from the 2000-year climate spin-up simulation driven by repeated monthly SMB fields during 1982–1999 from SEMIC modelling outputs (Yue et al., 2021), driven with downscaled and bias-corrected climate forcings by four Earth System Models. The equilibrium volume is slightly different than present day by -1.3% for BNU-ESM, -0.5% for HadGEM2-ES, and 0.8% for both MIROC models. Subplots (b–e) are the spatial distribution of VIC thickness differences (ice thickness after spin-up minus present ice thickness) from PISM driven by (from b–e) BNU-ESM, HadGEM2-ES, MIROC-ESM and MIROC-ESM-CHEM. The black curves represent the present ice cap extent. The magenta curves represent the extent after spin-up.

[Figure]

**Figure 4.** Top: Mean surface velocity over VIC from Sentinel-1, 100 m spatial resolution product (Wuite et al., 2021). Middle row: mean 2015–2020 surface velocities simulated by PISM under the RCP4.5 scenario from the 4 Earth System Model as labeled. Bottom row: the PISM-Sentinel differences.

[Figure]

**Figure 6.** The ice thickness differences from PISM outputs between the year 1982 and 2020 over Vatnajökull ice cap under G4 (1st row), RCP4.5 (2nd row) and RCP8.5 (3rd row) scenarios, and their differences (G4-RCP4.5, 4th row; G4-RCP8.5, 5th row) by Earth System Model (ESM, from left to right), BNU-ESM, HadGEM2-ES, MIROC-ESM, MIROC-ESM-CHEM and ensemble mean. The initial state in 1982 is from each separate ESM.

[Figure]

**Figure 7.** a) Ensemble mean of ice cap height differences, 2089 minus 1982 caused by SMB ([1st] row) and non-SMB (i.e., ice dynamics and basal melting, ([2nd] row)) calculated as the difference between MB and SMB, under the G4, RCP4.5 and RCP8.5 scenarios. No change is marked by the dashed black curves. b) Ensemble mean differences (G4-RCP4.5 and G4-RCP8.5) in ice cap thickness by 2089 due to SMB (1st row) and ice dynamics (2nd row). c) Decadal ensemble means of modelled mass balance (solid curves), SMB (dashed curves) and non-SMB (dotted curves) under historical (magenta), G4 (red), RCP4.5 (blue) and RCP8.5 (black) scenarios. The vertical lines denote the beginning and the end of SAI geoengineering. Individual models are in Figs. S2–S5.

38: How do you measure surface MASS balance in meters per year? Is this meters per year of ice? If so, what ice density do you assume? Or is it m/yr of "ice water equivalent?" In which case, why not just stick with mass units and call it 800 kg/m^2? Or is it decrease in elevation with unknown density?

The SMB unit is 'ice water equivalent'. We rewrite the sentence in Section 2.1:

"SEMIC (Surface Energy and Mass balance of Intermediate Complexity; Krapp et al., 2017) is a surface energy and mass balance model that is capable to estimate ice sheet/glacier SMB (unit: water equivalent m yr$^{-1}$)."

51: strike "ever" and "very". Don't use the word "very", ever.

OK. We corrected:

"here we focus only on impacts from SAI on the mass balance of a single ice cap in Iceland. The topic is of wider interest because the behaviour of the North Atlantic under both climate models driven by greenhouse gases, and observational evidence points to a slow-down in AMOC, leading to a much-reduced rate of warming in the North Atlantic relative to the rest of world (Cheng et al., 2013)."

52: How is behavior of North Atlantic atypical? Is it warmer or colder than "typical" and why? "because of the AMOC" is a bit of a mysterious reason.

OK. We corrected:

"Although there are many other potential impacts that might be expected if SAI were undertaken, here we focus only on impacts from SAI on the mass balance of a single ice cap in Iceland. The topic is of wider interest because the behaviour of the North Atlantic under both climate models driven by greenhouse gases, and observational evidence points to a slow-down in AMOC, leading to a much-reduced rate of warming in the North Atlantic relative to the rest of world (Cheng et al., 2013). Under SAI, AMOC slows less than under greenhouse gas climates (Hong et al., 2017; Yue et al., 2021; Xie et al., 2022). Thus, in Iceland, we would expect SAI changes on AMOC and radiative forcing to have compensatory effects to the ice cap. Furthermore, the Arctic warmed 6 times faster than the global mean from 1998-2012 (Huang et al., 2017), leading to concerns on the stability of the Arctic cryosphere, and examination of possible roles for geoengineering methods in its preservation (Lee et al., 2021).Whether SAI might even lead to exacerbated ice mass loss in the North Atlantic is an important question that goes to the fundamental reason for ever doing SAI – that is does SAI better preserve the important elements of the current climate system than plausible greenhouse gas emissions scenarios?"

54: What do words like "considerably" and "much less" mean??

'considerably' means the AMOC slows sharply. 'much less' means the degree of AMOC weakens in future under geoengineering (Hone et al., 2017; Yue et al., 2021) less than without geoengineering. We rewrite:

"Thus, in Iceland, we would expect SAI changes on AMOC and radiative forcing to have compensatory effects to the ice cap. Furthermore, the Arctic warmed 6 times faster than the global mean from 1998-2012 (Huang et al., 2017), leading to concerns on the stability of the Arctic cryosphere, and examination of possible roles for geoengineering methods in its preservation (Lee et al., 2021).Whether SAI might even lead to exacerbated ice mass loss in the North Atlantic is an important question that goes to the fundamental reason for ever doing SAI – that is does SAI better preserve the important elements of the current climate system than plausible greenhouse gas emissions scenarios?"

55: "Furthermore, polar amplification leads to concerns on the stability..."; and add a reference for polar amplification.

We rewrite:

"Furthermore, the Arctic warmed 6 times faster than the global mean from 1998-2012 (Huang et al., 2017)"

57: Langue seems overly wordy

We rewrite:

"Furthermore, the Arctic warmed 6 times faster than the global mean from 1998-2012 (Huang et al., 2017), leading to concerns on the stability of the Arctic cryosphere, and examination of possible roles for geoengineering methods in its preservation (Lee et al., 2021)"

58: Not sure what effects are unwelcome? The descriptions of geoengineering effects earlier in the paragraph seem welcome. I'm confused...

Unwelcome impacts mean the geoengineering may fail in this region due to the Arctic amplification and the impact of enhanced AMOC under geoengineering. We changed to:

"Although there are many other potential impacts that might be expected if SAI were undertaken, here we focus only on impacts from SAI on the mass balance of a single ice cap in Iceland. The topic is of wider interest because the behaviour of the North Atlantic under both climate models driven by greenhouse gases, and observational evidence points to a slow-down in AMOC, leading to a much-reduced rate of warming in the North Atlantic relative to the rest of world (Cheng et al., 2013). Under SAI, AMOC slows less than under greenhouse gas climates (Hong et al., 2017; Yue et al., 2021; Xie et al., 2022). Thus, in Iceland, we would expect SAI changes on AMOC and radiative forcing to have compensatory effects to the ice cap. Furthermore, the Arctic warmed 6 times faster than the global mean from 1998-2012 (Huang et al., 2017), leading to concerns on the stability of the Arctic cryosphere, and examination of possible roles for geoengineering methods in its preservation (Lee et al., 2021).Whether SAI might even lead to exacerbated ice mass loss in the North Atlantic is an important question that goes to the fundamental reason for ever doing SAI – that is does SAI better preserve the important elements of the current climate system than plausible greenhouse gas emissions scenarios?"

58: "Effects" is a fine word, no reason to use "impacts." Impact is a verb when one body hits another and makes a crater... Turning that into a noun and using instead of an existing good word ("effects") is torturing our poor English language.

'impact' can be a noun, and the usage is as popular as verb. See oxford English dictionary: https://www.oed.com/search?searchType=dictionary&q=impact&_searchBtn=Search

And see IPCC report https://www.ipcc.ch/report/ar6/wg2/ "The Working Group II contribution to the IPCC Sixth Assessment Report assesses the impacts of climate change, looking at ecosystems, biodiversity, and human communities at global and regional levels."

60: strike "state-of-the-art." This is a science paper, not marketing material. Instead of "state-of-the-art," just state which version of PISM you used. eg; PISM 3.5 (or whatever).

Ok. We changed:

"We simulate the response of the VIC with the Parallel Ice Sheet Model (PISM; version 1.0) driven by

monthly SMB from 1982–2089 under CMIP5 historical (1982–2005), RCP4.5 (2006–2089), RCP8.5 (2006–2089) and SAI G4 (2020–2089) scenarios."

63: Just say **"business as usual" scenario**. I understand it's a scenario we won't like. But it's an oxymoron to label "business-as-usual" as "extreme", especially if it's the most likely scenario. In other words, your use of the word "extreme" is poorly defined here.

No, we disagree. The emissions commitments given by the countries that have signed up to the Paris 2015 accord produce a trajectory closest to the RCP4.5 scenario (Kitous and Keramidas, 2015). Countries in general follow international agreements and commitments because the alternative is chaos. Hence RCP4.5 is the most likely scenario. It is precisely because countries have recognized that "business as usual" will be extremely costly and have decided to cap their emissions. Business as usual is an extreme because it assumes no mitigation, and just about every country accepts the need for mitigation, the question is how much will be done. Thus, business as usual bounds one end of the emissions trajectories, i.e. it is an extreme. However, we rewrite this sentence:

"RCP4.5 (Thomson et al., 2011) is a stabilization scenario with emissions similar to those agreed under the Paris 2015 agreement (Kitous and Keramidas, 2015), while RCP8.5 (Riahi et al., 2011) is a "business-as-usual" scenario that is a likely outcome if we do not make any efforts to reduce the greenhouse gas emissions."

63: Not sure of the word "branches off? Maybe use "depart" instead? Anyway, why are we using a scenario that started in 2020, when it's now 2022?

Branching off is the standard terminology used in climate modeling for beginning a new scenario from an existing one. E.G. https://agupubs.onlinelibrary.wiley.com/doi/10.1029/2019MS002009 and countless other articles. G4 has been run by various climate models. There is a wealth of data available to analyze. This article is, we thought, clearly an academic exercise to examine possible impacts of SAI on ice in the North Atlantic. It is not an operational forecast. There are no significant differences in climate across scenarios in the first few decades of the 21$^{st}$ century, it makes no practical difference that we are in the year 2022 already.

67: Did you really use PISM version 1.0? Please explain which vesrion of PISM you ACTUALLY used. Also, please use correct PISM URL: https://www.pism.io

Yes, we do use PISM 1.0, although the newest version is 2.0.2. We changed the URL as your corrections.

72: Are they "free parameters" or "hyperparemeters"? Are the two terms different, and which is more appropriate here?

Sorry we don't understand this question as we have never heard the term "hyperparemeters". According to google they exist in relation to machine learning, which has no relevance here. They are parameters in the sense that we understand the meaning of the word parameter. We are using it in the standard way.

73-74: Do you have references for any of these schemes?

We added references and the description for each parameterization.

The PISM model (version 1.0; Bueler and Brown (2009); https://www.pism.io) is an open-source ice sheet thermo-dynamic model that has been used in numerous studies of a wide range of ice sheets and glaciers (e.g., Aschwanden et al., 2019; Yan et al., 2020). The evolution of the ice cap surface elevation

$H$ is calculated by mass continuity equation:

$$\frac{dH}{dt} = M - \nabla \cdot \vec{Q} - M_b, \tag{2}$$

where $t$ is the time step, $M$ is the mass balance, $M_b$ is the basal melt rate, $\nabla \cdot \vec{Q}$ is the ice flux calculated by stress balance model. PISM provides several parameterizations to describe the ice stress balance, ice flow, basal sliding and ice calving (details see PISM manual; https://www.pism.io/docs/). The choices of parameterizations and free parameters followed Schmidt et al. (2020), and validated the simulations using observations over Vatnajökull. In brief the parameterizations we used in this study are:

—We use hybrid stress balance model (Bueler and Brown, 2009) with both Shallow Ice Approximation (SIA; Hutter, 1984) and Shallow Shelf Approximation (SSA; Morland, 1987) to solve ice vertical deformation and longitudinal stretching, allowing simulation of both slowly flowing ice cap interiors and fast flowing outlet glaciers.

—Ice rheology is parameterized by Glen's flow law (Glen, 1955):

$$\tau = 2\eta D \,, \tag{3}$$

where $\tau$ is the deviatoric stress tensor, $D$ is the strain rate tensor, and $\eta$ is given by:

$$\eta = \frac{1}{2} A(T)^{-1/n} d_e^{(1-n)/n}, \tag{4}$$

where the parameter $A$ is strongly dependent on ice temperature, $d_e$ is the second invariant of the strain rate tensor, flow exponent $n$ is commonly taken the value of 3.

—Ice front calving rate $c$ is calculated by Eigen scheme (Levermann et al., 2012):

$$c = K \cdot \max(0, \epsilon_{\parallel}) \cdot \max(0, \epsilon_{\perp}) \,, \tag{5}$$

where $K$ is a constant that explains the ice properties relevant to calving, $\epsilon_{\parallel}$ and $\epsilon_{\perp}$ denote the strain rate along and transversal to horizontal ice flow, respectively.

—Basal sliding is estimated by a pseudo-plastic law (see PISM manual; https://www.pism.io/docs/), which estimates the basal shear stress $\tau_b$ through the yield stress $\tau_c$, basal velocity $u_b$, and the parameters of velocity threshold $u_{threshold}$ and power $q$:

$$\tau_b = -\tau_c \frac{u_b}{u_{threshold}^q |u_b|^{1-q}}, \tag{6}$$

To initialize PISM over VIC, we need surface elevation, bedrock altitude, upward geothermal flux, ice temperature, and monthly SMB for boundary conditions (Table 1, Fig. 1, Fig. 2). We re-gridded these data to a 500 × 500 m resolution in order to balance the computation costs and a more realistic ice surface velocity. As with Schmidt et al. (2020), we prescribe temperate ice at 0°C for the whole VIC, as Icelandic glaciers are temperate (Björnsson and Pálsson, 2008). Upward geothermal flux is from Flowers et al. (2003) and assigns a value of 0.18 W m$^{-2}$ in the eastern sector of VIC, zero in the western sector, and 50 W m$^{-2}$ for the Grímsvötn active volcano (Björnsson, 1988). Surface elevation is from observations by the Spot5 satellite (Berthier and Toutin, 2008), which represents the period from June to September in 2010. Bedrock topography is based on radio echo profiles since 1980 (Björnsson, 1986, Björnsson and Pálsson, 2020).

Ok. We made a table to be clearer.

**Table 1** A summary input data fields in PISM.

| PISM input fields | Data source | Period | PISM running resolution | Reference |
|---|---|---|---|---|
| Surface mass balance | SEMIC output driven by downscaled and bias-corrected climate fields from [a] BNU-ESM, [b] HadGEM2-ES, [c] MIROC-ESM, [d] MIROC-ESM-CHEM | 1982–1999, repeated for 2000 years (PISM spin-up) 1982–2005 (CMIP5 historical) 2006–2089 (RCP4.5, RCP8.5) 2020–2089 (GeoMIP G4) | Monthly; 500 ×500 m | Yue et al. (2021) |
| Surface elevation | Spot5 satellite | June to September 2010 | 500 ×500 m | Berthier and Toutin. (2008) |
| Bedrock topography | Radio echo profiles | 1980 | 500 ×500 m | Björnsson, (1986); Björnsson and Pálsson. (2020) |
| Ice cap thickness | Surface elevation minus bedrock topography | — | 500 ×500 m | — |
| Upward heat flux | Assigns typical values | — | 500 ×500 m | Flowers et al. (2003); Björnsson. (1988) |
| Ice temperature | Prescribed 0 °C everywhere | — | 500 ×500 m | Schmidt et al. (2020) |

[a] Ji et al. (2014), [b] Collins et al. (2011), [c,d] Watanabe et al. (2011).

It's about 1.2 km. We added the km resolution: 0.025°×0.025° (about 1.2 km×1.2 km).

Figure 1:

i.      (d) and (e) should be reversed; as generally things are laid out from left-to-right, not in circular or spiral patterns.

Actually often things are laid out in clockwise or anti-clockwise directions. But we changed them as wished.

ii.      Use standard terminology such as ELA, instead of "equilibrium line boundary."

ok

iii.      For (e), why is it so artificial?

We assume you mean why is there a sudden jump in geothermal heat at the volcanic hot spot? The temperature scale is not smooth or linear (see your other comments).

There are no heat flux observations at the Vatnajökull, so we followed settings from Flower et al. (2013), we set 0 (western sector, subsurface hydrothermal circulation is believed to be sufficiently vigorous to prevent heat from reaching the base of the ice), 0.18 W m$^{-2}$ (eastern sector), and 50 W m$^{-2}$ (volcanic hot spot).

iv. Use a better color scale, i.e. (a) one made for elevations, and (b) a scale with discrete segments, not continuous.   It's hard to tell what's what with contiguous scales.   See here for color schemes suitable for topography: http://soliton.vm.bytemark.co.uk/pub/cpt-city/views/topo.html

http://soliton.vm.bytemark.co.uk/pub/cpt-city/ncl/tn/topo_15lev.png.index.html

ok

iv.   In camption, but the Berthier & Toutin reference AFTER "in summer 2010"

This is the correct place since we cite a methods paper for the SPOT5, nothing to do with Iceland. The data for Iceland has not been published, so we only cited the methods. We changed:

"**Figure 1.** Model input data fields. (a) Vatnajökull ice cap (VIC) surface elevation from Spot5 (data processing methods see Berthier and Toutin, 2008) in summer 2010;"

vi. For (d), use a +/- gradient color scale (again, with discrete colors not continuous).  Blue/red is good; with blue as SMB>0 and red as SMB<0.  This is a good one:

http://soliton.vm.bytemark.co.uk/pub/cpt-city/gery/tn/seismic.png.index.html

http://soliton.vm.bytemark.co.uk/pub/cpt-city/ncl/tn/temp_19lev.png.index.html

With a properly centered color palette, you won't need to draw the ELA as a dotted line.

Thanks for your suggestions. We revised Figure 1:

[Figure]

**Figure 1.** Model input data fields. (a) Vatnajökull ice cap (VIC) surface elevation from Spot5 (data processing methods see Berthier and Toutin, 2008) in summer 2010; (b) bedrock elevation (Björnsson, 1986; Björnsson and Pálsson.2020); (c) ice thickness; (d) applied upward geothermal heat flux (Flowers et al. 2003), including the Grímsvötn active volcano. (e) annual average surface mass balance 1982-1999 simulated by SEMIC forced by four Earth System Models (Yue et al., 2021). (f) the geographical location of panel (a, red box) observed by Google Earth.

93: Why spinup for 200 years?  Figure 2 seems to suggest 400 is sufficient. Or why not 4000 years?
Not exactly right. If you carefully look at volume in 400 years and 2000 years, it does show difference!! (about 15 km$^3$). Actually, when spin-up after 1500-year, volume shows little change, so no need to extend to 4000 years.